

# Between a rock and a hard place: experimental assessment of recruitment patterns in a bathyal environment of the Low Arctic

Sophie Wolvin[1], Jean-François Hamel[2] and Annie Mercier[1]

[1] Memorial University of Newfoundland, St. John's, Canada
[2] Society for the Exploration and Valuing of the Environment (SEVE), St. Philips, Canada

## ABSTRACT

The study of larval transport and recruitment in the deep sea is crucial to the understanding of species distributions, community assembly, and the potential effects of anthropogenic activity and climate change on the maintenance of biodiversity. This study sought to better understand the role of substratum features in deep-sea larval recruitment at high latitudes. Four settlement frames composed of blocks of different substrata (mesh, plastic, stone, and wood) were deployed for 9 to 13 months at bathyal depths in the Labrador Sea (northeastern Canada). Colonial hydrozoans dominated as recruits, with one taxon (family Campanulariidae) colonizing all substratum types across all geographic sites. Other taxa, including arthropods, octocorals, and other anthozoans recruited only onto specific substrata and consistent microhabitats within them. Overall, higher morphospecies and phylum richness characterized the three-dimensional mesh substratum relative to other substratum types, whereas the complex surface area offered by projections in the plastic substratum had higher densities of individuals or colonies for all morphospecies combined. Wood, offered as a single elongated panel, was the most heavily colonized, whereas both mesh and stone hosted morphospecies not found on any other substratum type. Factors such as geographic location, depth, altitude above the sea floor, and orientation/obstruction of the frame, may have modulated recruitment patterns. These results provide foundational knowledge on larval recruitment patterns and early colonization by opportunistic hard-bottom benthic taxa in a poorly-studied region of the Eastern Canadian deep sea.

# INTRODUCTION

Documentation of early successional stages of deep-sea communities on hard substrata is limited for polar regions, in part because of their remoteness and the research challenges associated with severe fall and winter meteorological conditions and seasonal ice cover. However, some investigations have examined early recruitment and succession in shallow-water benthic environments, where ice-scour regularly impacts communities

Corresponding author
Sophie Wolvin, swolvin@mun.ca

(*Dayton, 1989*; *Stanwell-Smith & Barnes, 1997*; *Brown et al., 2004*; *Barnes & Kukliński, 2005*; *Bowden, 2005*; *Bowden et al., 2006*; *Konar, 2007*, *2013*; *Kukliński et al., 2013*; *Al-Habahbeh et al., 2020*). They have shown a marked recruitment seasonality in both the Arctic and Antarctic (*Bowden, 2005*; *Kukliński et al., 2013*; *Meyer et al., 2017*), with slow colonization rates of new substrata extending years to decades (*Stanwell-Smith & Barnes, 1997*; *Brown et al., 2004*; *Konar, 2007*; *Konar, 2013*; *Al-Habahbeh et al., 2020*). Whereas species richness in the shallows of polar regions sometimes resembles that of temperate regions (*Barnes & Kukliński, 2005*), local habitat, depth (experimental ranges from 8–200 m), and distance between similar habitats heavily influence recruitment at high latitudes (*Bowden et al., 2006*; *Barnes, 2017*; *Meyer et al., 2017*).

Sessile and sedentary species dominate hard-bottom marine communities worldwide, mostly settling on the substratum as planktonic larvae after a period of epibenthic exploration at the end of their pelagic life (reviewed in *Jenkins, Marshall & Fraschetti (2009)*). Recruitment patterns on a specific substratum may relate to the nature, texture or roughness of a surface (*Walters & Wethey, 1996*; *Gilg et al., 2010*; *Sun, Hamel & Mercier, 2010*, *2011*; *Meyer, Li & Young, 2018*), chemical cues emanating from the biofilm (*Morse et al., 1996*; *Sun, Hamel & Mercier, 2010*; *Hadfield, 2011*), and the presence of conspecifics for certain gregarious species (*Pawlik, 1986*; *Johnson & Woollacott, 2010*). Other confounding factors influence recruitment rates and patterns, including regional variation in larval supply and dispersal rates, direction or strength of water flow, and rates of post-settlement mortality resulting from predation, competition, physiological stress, and physical or biological disturbances (*Gaines & Bertness, 1992*; *Hunt & Scheibling, 1997*; *Palardy & Witman, 2013*; *Hilário et al., 2015*; *Guy & Metaxas, 2022*). In the deep sea, spatial fragmentation of communities can complicate population connectivity and larval dispersal (*Hilário et al., 2015*).

Recruitment studies often use deployments of replicable settlement frames, arrays, or "collectors" to mimic a range of hard substrata available on the ocean floor to examine the appearance of pioneer species (reviewed in *Davis (2009)*). Several studies have used such arrays, especially in shallow temperate and tropical coral reefs (*Chalmer, 1982*; *Jenkins, Marshall & Fraschetti, 2009*) and coastal waters (*Walters & Wethey, 1996*; *Migotto, Marques & Flynn, 2001*; *Denitto, Terlizzi & Belmonte, 2007*; *Gilg et al., 2010*). Studies may offer mixed blocks or panels of natural substrata, *e.g.*, basalt, wood, glass-sponge fragments (*Beaulieu, 2001*; *Cuvelier et al., 2014*), as well as plastic and other synthetic surfaces (*Girard, Lacharité & Metaxas, 2016*; *Meyer-Kaiser et al., 2019*) representing increasingly common anthropogenic contaminants. The inclusion of variable surficial or internal complexities was important, in that species can exhibit selective preference towards certain substratum features or "microhabitats" (*Dumont, Gaymer & Thiel, 2011*). Larvae may settle on substrata that provide crevices or other shelters, and recruits then expand outward to colonize unsheltered surfaces (*Walters & Wethey, 1996*). Fluctuations in small-scale hydrodynamics due to different surficial features may also drive larvae to settle in a specific microhabitat over another (*Mullineaux & Garland, 1993*). Microhabitat heterogeneity has been suggested to enhance recruitment (*Barnes & Kukliński, 2005*). In addition, the features and topography of substratum types can influence further

colonization, as relative height of competing recruits can influence dominance in community hierarchies (*Walters & Wethey, 1986*). Despite their popularity in benthic studies globally and in shallow polar environments (*Teichert et al., 2012*; *Wisshak et al., 2022*), few studies have used settlement frames or similar apparatuses with checkerboard substratum designs to characterize larval recruitment patterns to deep-sea habitats.

In the Gulf of Maine (western North Atlantic), *Lacharité & Metaxas (2013)* deployed larval collector arrays composed of mosaic basalt rock plates and mesh pads and identified that the cold-water coral *Primnoa resedaeformis* colonized both substratum types, with greater abundance on more structurally complex portions of the frame than on the flat surfaces of the collectors. Moreover, *Girard, Lacharité & Metaxas (2016)* used these larval collectors to compare colonization of the two substratum types, and reported both higher diversity in species assemblages on the more complex (mesh) substratum and distinct clustering of species assemblages by substratum type. More recently, in another study in the Gulf of Maine, examination of video transects found peak recruitment of both *P. resedaeformis* and *P. arborea* at depths below 500 m, and dense aggregations of the glass sponge *Vazella pourtalesi* at 220–320 m (*Guy & Metaxas, 2022*), suggesting that small-scale environmental conditions, post-settlement processes, and supply of larvae play important roles.

In the Arctic regions, *Meyer-Kaiser et al. (2019)* noted that settlement on panels deployed for 19 years on a scientific lander at 2,467 m in the Fram Strait (between Greenland and Svalbard, Norway) was higher on brick than plastic substrata, and found species-specific preferences for panel altitude above the sea floor. In the same region, *Meyer-Kaiser, Plowman & Soltwedel (2021)* showed opportunistic recruitment of the otherwise rare hydrozoan *Boullonia cornucopia* on polycarbonate plastic panels, suggesting certain substratum types may promote its settlement. More recently, *Meyer-Kaiser et al. (2022)* compared early recruitment in shallow and deep-sea habitats of Atlantic and Arctic waters of the Fram Strait, and found that species composition differed mainly between the Atlantic and Arctic water masses rather than depth, with higher species richness in panels from the Atlantic. Moreover, the hydrozoans *B. cornucopia* and *Halisophonia arctica* tended to dominate at all depths (*Meyer-Kaiser et al., 2022*).

The present study sought to expand on the current understanding of benthic communities in the Labrador Sea (Northwestern North Atlantic) and on general recruitment patterns across differing substratum types. The Labrador Sea harbours many commercially valuable species such as Atlantic cod, herring, redfish, and snow crab (*DFO, 2021*). Many of these species utilize hard-bottom communities of the deep sea as nursery grounds (*Metaxas & Davis, 2005*; *Baillon et al., 2012*; *Pierrejean et al., 2020*; *DFO, 2021*), driving a need for a better understanding of factors that might promote the establishment of early-successional communities. Furthermore, previous experimental assessments of recruitment patterns in the deep sea at northern latitudes have tested only one or two substratum types. This is the first study to rely on a checkerboard multiple-choice design to maximize larval recruitment while exploring substratum colonization patterns in deep water at high latitudes. The checkerboard design allowed a first understanding of the fine capacity of larvae to choose precise type of bottom (including material, texture, crevices,
*etc.*). Standardized recruitment frames containing replicated substratum blocks of varying surficial and internal complexity (mesh, plastic, stone, and wood) were deployed for about a year (9 to 13 months) at each of four sites in the bathyal zone of the northern Labrador Sea (northeastern Canada) between 2017 and 2020. Species richness and abundance, and recruit size were analyzed with the goal of addressing two hypotheses: (1) whether substratum characteristics, including location on the surface and microhabitat complexity, drive community composition, and (2) whether recruitment patterns vary across geographic sites. This study also aimed to compare recruitment metrics across functional taxa (*e.g.*, unitary *vs.* colonial forms).

## METHODS

### Settlement frames

The settlement frames were built following the specifications used in the INDEEP project (www.indeep-project.org) designed in conjunction with project SERPENT (Scientific and Environmental ROV Partnership using Existing Industrial Technology) and Transocean to maximize recruitment (*Gates et al., 2017*; *Metaxas, Ramirez-Llodra & Hilário, 2022*). They were composed of three replicates of three different substratum types in a standardized block shape: folded pads of mineral and synthetic fibres (Scotch-Brite™), hereafter called "mesh"; interlocking plastic blocks (DIMPLE™ Bristle Stacking Blocks), hereafter called "plastic"; and blocks of limestone (calcium carbonate), hereafter called "stone". Each block measured approximately $5 \times 5 \times 5$ cm and was bolted horizontally through the center to a fiberglass frame in a randomized $3 \times 3$ checkerboard grid, where four faces were fully exposed and two opposing faces contained the bolt hole by which it was attached to the frame (Fig. 1). A fourth substratum type, a single piece of pine wood measuring $5 \times 5 \times 20$ cm and hereafter called "wood", was bolted externally to one side of the frame. Each substratum block (except wood) was made up of six faces of equal size ($10$ cm$^2$ each), which for this study were additionally subdivided into recruitment locations: edge or centre (Fig. 1A, Table S1) and sheltered or unsheltered surface features or "microhabitats" (Fig. 1B, Table S2). The wood panel had five unequal exposed surfaces (three faces of $100$ cm$^2$ and two faces of $10$ cm$^2$) (Fig. 1).

### Deployment method and sites

Six settlement frames were deployed initially on either moorings or landers (Fig. 2); four were recovered successfully (at Sites 1, 2, 3, and 4) while fatal corrosion of mooring anchors resulted is loss of two before they could be retrieved (Sites 5 and 6). All frames were deployed from the icebreaker CCGS *Amundsen* in the Labrador Sea, approximately 170 km offshore of the northernmost tip of Labrador, Canada, at depths between 400–1,000 m (Fig. 2; Table S3). Deployments occurred from 2017 to 2021, and the frames remained in the water for a period of 9 to 13 months (Table S3). The Site 1 frame was deployed on a mooring to 499 m depth at 11 m altitude (*i.e.*, height above sea floor). The Site 2 frame was deployed on a mooring to 960 m depth and 60 m altitude, approximately 6 km from Site 1. The Site 3 frame was deployed on a lander to 409 m depth and 1 m altitude, approximately 2 km from Site 1 and 7 km from Site 2. The Site 4 frame was

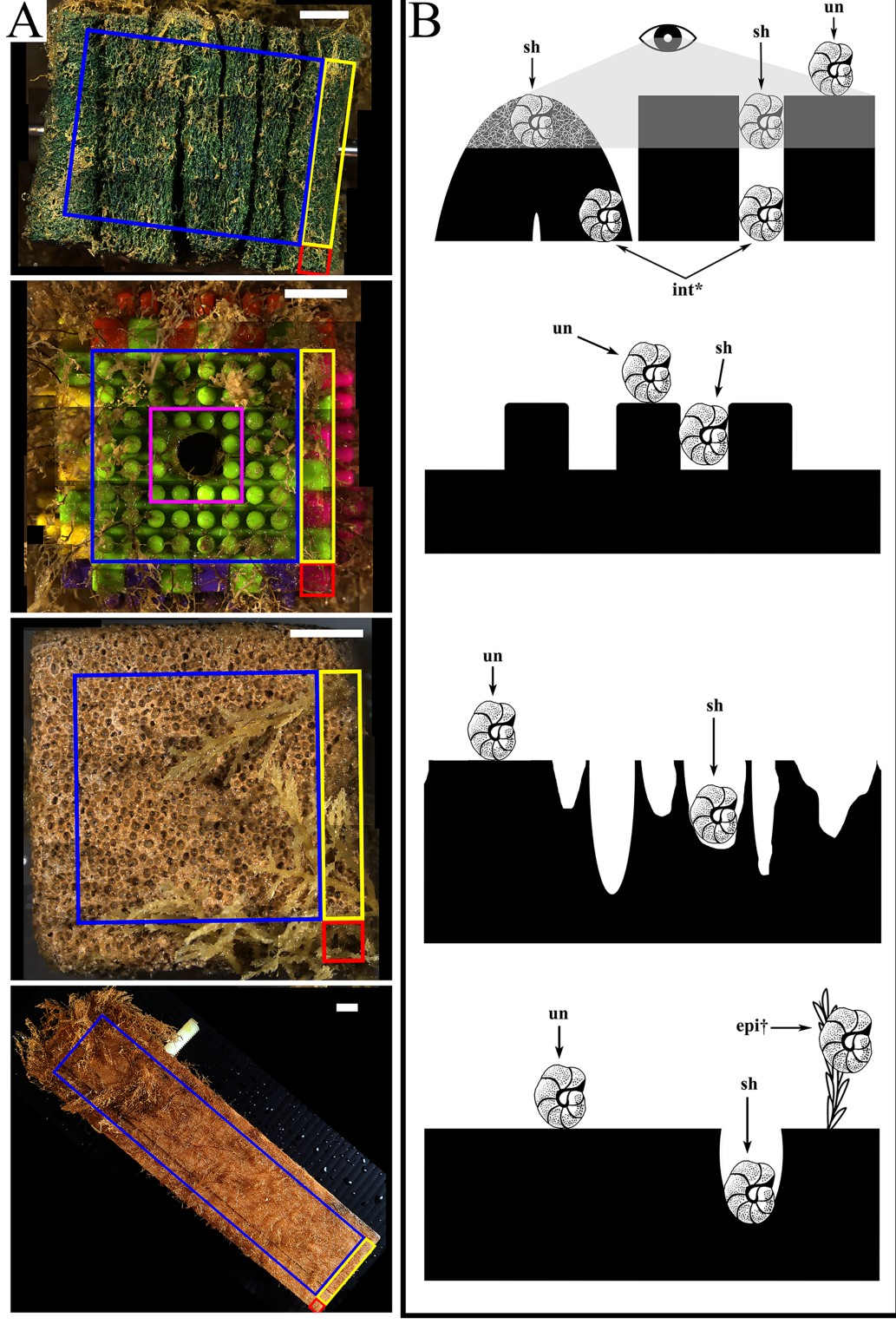

**Figure 1 The four substratum types deployed at each geographic site with illustration of microhabitats.** (A) For all substratum types, recruitment locations used during this study are highlighted as "edge" (within 5 mm of any meeting point between 2 or 3 faces): including corners (red) and edges (yellow); and centres (blue). Excluded were bolt areas (pink), with the white arrows indicating the bolt hole through which the substratum block was attached. From top to bottom: mesh, plastic, stone, and

**Figure 1** (continued)
wood recruitment locations analysed. (B) For all substratum types, recruitment microhabitats used are indicated with arrows as sheltered (sh) and unsheltered (un) and the diagram scale is 1 cm. Recruitment as epibionts (epi) and internal (int) were excluded from microhabitat calculations. From top to bottom: mesh (grey area indicates visible field), plastic, stone, and wood microhabitats analyzed. All scale bars represent 10 mm. Diagrams further defined in Tables S1, S2.

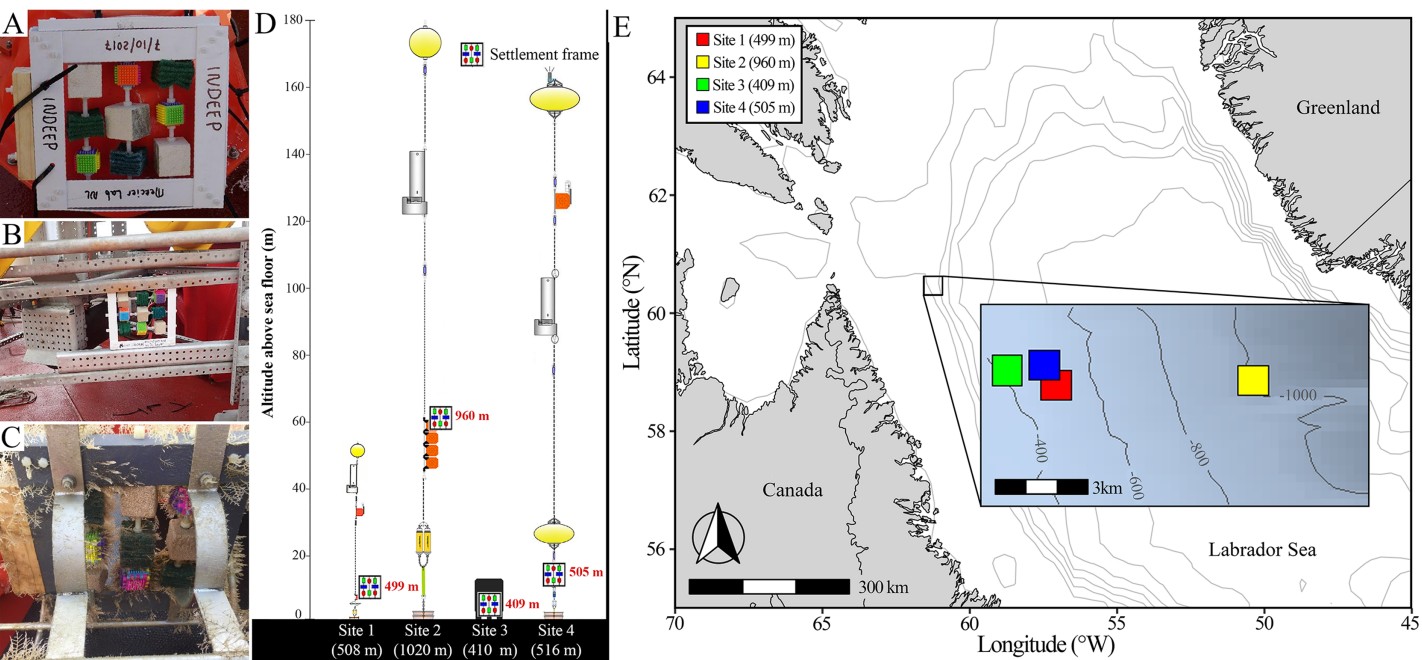

**Figure 2 Deployment and geographic location of settlement frames.** (A) Pre-deployment settlement frame, showing the standardized distribution of substratum blocks, deployed attached to a buoy on a scientific mooring as in Site 1 and 2. (B) Pre-deployment settlement frame deployed on a lander as in Site 3. (C) Post-retrieval settlement frame deployed in an open cage on a scientific mooring as in Site 4. (D) Scientific mooring and lander deployment diagrams showing position of the settlement frame on the apparatus, altitude above bottom, and its approximate depth in red, with the site name and total depth of sea floor below each deployment apparatus diagram in white (diagrams courtesy of Shawn Meredyk, Amundsen Science, and ArcticNet). (E) Map of the deployment locations of the four successfully retrieved settlement frames. Exact location, depth, and altitude above the sea floor details can be found in Table S3.

deployed on a mooring at 505 m depth and 11 m altitude, approximately 2 km from both Sites 1 and 3, and 6 km from Site 2 (Fig. 1). As a result of logistical constraints, frames at Sites 1 and 2 were laid flat on the mooring or lander apparatus (*i.e.*, obstructed and limited flow through the frame), whereas frames at Sites 3 and 4 were attached from the side (*i.e.*, unobstructed and allowing full flow through the frame) (Fig. 2).

At recovery, settlement frames were removed and disassembled from the supporting apparatus and immediately preserved substrata either in 100% ethanol (all mesh, plastic, and stone blocks unless otherwise noted) or frozen (all wood panels, and the entire frame from Site 3). The wood block from Site 3 was lost during recovery. Samples initially preserved by freezing were transferred to 100% ethanol prior to analysis. All preserved samples were transported to and analyzed at the Ocean Sciences Centre of Memorial University (Newfoundland and Labrador).

## Data collection and analysis

Mosaic images of each face of each substratum block were generated using a Leica M205 stereo microscope and the Leica Application Suite X (LAS X) Life Science Microscope Software Platform, which were then stitched together in Adobe Photoshop to complement direct analysis of each substratum surface under the stereo microscope. In the case of the wood block, a diagram was used in place of a mosaic image. To provide a spatial reference, a grid composed of 1 × 1 cm squares was overlaid digitally on the mosaic image (Fig. S1). The mosaic images were used to map the locations and percent cover of each individual/ colony of each species, and the grid overlay was used for digital analysis of species abundance (described below).

*Identification.* Because most colonizers observed on the various substrata were juveniles (often not showing the taxonomic characters required for identification to species), all were assigned to the lowest taxonomic level possible and to a morphospecies (*e.g.,* "*Eudendrium* msp. 1"). Each morphospecies (msp/mspp) was also categorized as either unitary or colonial, defining a colony as any biogenic structure that connected down to a single base (excluding horizontal stolonization). Both singletons and colonies were scored and referred to as "individuals". Morphospecies were also characterised as motile (capable of movement away from initial location of larval recruitment, including sedentary taxa) or sessile (incapable of movement away from initial recruitment location). After photographing all individuals in the original place on the substratum blocks, they were detached and preserved in 100% ethanol.

*Reproductive status.* For each individual, ontogenetic stage was estimated and categorized as established (*e.g.,* adult unitary, or colony containing more than one module), juvenile (*e.g.,* identifiably juvenile or colony of just one module), or eggs. Reproductive individuals were defined as those with visible gametes/embryos (arthropods) or gonozooids (hydrozoans).

*Richness, biodiversity, and frequency of occurrence.* Richness was defined as the number of morphospecies present per substratum block, and phylum richness as the number of phyla per substratum block. Morphospecies richness within a phylum was also examined. Richness was examined both as a total for all blocks combined (sum of all morphospecies or phyla) as well as an average per substratum block (across the three substratum blocks of each type in each frame ± SD; except wood). Diversity was calculated as the Shannon Index (*Shannon, 1948*) using abundance of morphospecies:

$$H' = \sum_{i=1}^{s} (\rho_i) \ln \rho_i$$

where ρi is the proportion of individuals of one morphospecies divided by the total number of individuals, ln is the natural log, and s is the number of morphospecies.

Frequency of occurrence was defined as the percentage of substratum types or geographic sites a morphospecies recruited to out of the total number of substratum types or sites examined.

*Morphospecies abundance.* Abundance was calculated as the total number of individuals as well as individuals per substratum block (ind block$^{-1}$). Abundances of colonial or high-density unitary morphospecies (*e.g.*, colonial hydrozoans) were estimated as the total number of individuals counted within three randomly selected 1 cm$^2$ squares of a grid overlay on the mosaic image of each face (see above; Fig. S1), which were averaged and then extrapolated to the whole face (± SD). The eroded corners of all substratum blocks (as defined in Fig. 1 and Table S1) were excluded from the analysis of high-density or colonial morphospecies. At the level of the block, morphospecies abundance was calculated both as a sum of individuals (total abundance) and as the average number of individuals on each face of the block (abundance per block face) to include the variability of recruitment on different faces. In the case of abundance per block face, the least colonized face was omitted from all blocks to account for obstructed faces in Sites 1 and 2.

*Surface cover.* The proportion of the block surface occupied by a given morphospecies (or group of morphospecies) was calculated using estimated increments of 5% visually at two levels: (1) at the surface of the substratum (up to 1 mm height) and (2) at the canopy (over 1 mm; particularly for arborescent forms like colonial hydrozoans). Global and morphospecies-specific cover represented an average across the three blocks of each substratum type at each site (± SD; except wood).

*Spatial recruitment and colonization patterns.* How each morphospecies spatially recruited to and colonized the surface of the block was categorized in two ways: location and microhabitat. Recruitment location was established per morphospecies on each face of a given block by scoring its presence/absence at the edges or in the centre (Fig. 1; see details of the locations in Table S1). The following equation calculated how often a morphospecies occurred in each location (*i.e.*, number of block faces where present in each location) out of the total number of occurrences of each morphospecies (*i.e.*, the number of block faces on which it was present) as a percentage:

$$\%(y)\text{occurences} = \frac{\text{Total faces on (x) the morphospecies occurred in (y)}}{\text{Total faces on (x) the morphospecies occurred}} \times 100$$

where x is substratum type and y is the location. Morphospecies that occurred in multiple locations had each location counted separately, thus all occurrences in each location are accounted for.

Pores, projecting pegs, and folds characterized the stone, plastic and mesh substrata respectively, whereas colonizers could bore into wood. To consider this three-dimensional aspect, these features were considered to be "sheltered" microhabitats, in contrast to the "unsheltered" remainder of the surface (*i.e.*, outermost surface area around pores on stone or folds in mesh, and the flat tops of pegs on plastic) (Fig. 1; see details of the various microhabitats in Table S2). To examine the occurrences of colonizers in each microhabitat of the total number of occurrences of each morphospecies, the same equation as described above for recruitment location was used, with x as substratum type and y as microhabitat category. Morphospecies that occurred as epibionts (*i.e.*, not touching any part of the

substratum block) were categorized separately and excluded from the location and microhabitat calculations (Fig. 1B).

*Effect of substratum and geographic site.* The effect of substratum type (sites pooled) and geographic site (substrata pooled) were both assessed for each metric defined above.

*Epibiosis.* Any epibiotic pairings (*i.e.*, one basibiont and one epibiont) present were documented opportunistically for observations on succession in early communities. Richness and abundance measurements included epibiota, defined as individuals that occurred on other individuals, but were included only in the canopy for percent cover measurements.

## Statistical analyses

Multivariate analyses used PRIMER v7 software. Differences in morphospecies abundance, density, base and canopy coverage, richness, and diversity (H') between substratum types and geographic sites were explored using PERMANOVA (unrestricted permutation of raw data; type III partial) on Bray-Curtis resemblance matrices and visualized with non-metric multidimensional scaling (nMDS). A two-way crossed analysis of similarity (ANOSIM) test with Spearman rank correlation compared between sites and substratum types. Density, coverage, richness, and diversity were square root transformed to balance between both the most common and rarer morphospecies; abundance values were fourth-root transformed as the wide range of values needed further balancing (*Clarke et al., 2014*).

# RESULTS

## Trends in abundance, richness, diversity, and coverage

Combining all substratum types and geographic sites yielded a total of 127,724 individuals representing 28 mspp across seven phyla, as well as three unidentifiable taxa (107 ind) that were excluded from further analyses unless otherwise stated. Overall, there were 1.8 ± 1.3 mspp per block with 25.0 ± 18.4% surface coverage and 22.1 ± 20.4% canopy coverage. This fauna included a mix of 11 colonial (127,191 ind) and 17 unitary mspp (426 ind), 17 of which were sessile (127,451 ind) and 11 of which were motile (165 ind).

Across geographic sites and settlement frames, the diversity of morphospecies was composed of nine cnidarians including one octocoral, one actiniarian, and seven colonial hydrozoans (Fig. 3A); one of the latter occurred in two forms, *i.e.*, an erect branching colony (Campanulariidae msp. 2A) and another of stolonate colony (Campanulariidae msp. 2B; *i.e.*, horizontal growth). There were also seven arthropods including one halacarid, one ostracod, one motile and one tube-dwelling gammarid amphipod, one caprellid amphipod, one isopod, and one copepod (Fig. 3E); four foraminifers (Fig. 3B); four poriferans (sponges; Fig. 3G); two annelids which included one free-living and one tube-dwelling polychaete (Fig. 3F); two molluscs which included one gastropod, and one gastropod egg mass (Fig. 3D); and one radiolarian (Fig. 3C). The three unknown mspp included individual eggs seen on multiple occasions, one egg mass, and one unidentifiable aggregate of biological origin (Fig. 3H). Table 1 details all morphospecies present and their occurrences. None occurred everywhere (*e.g.*, on all substratum types at a given site).

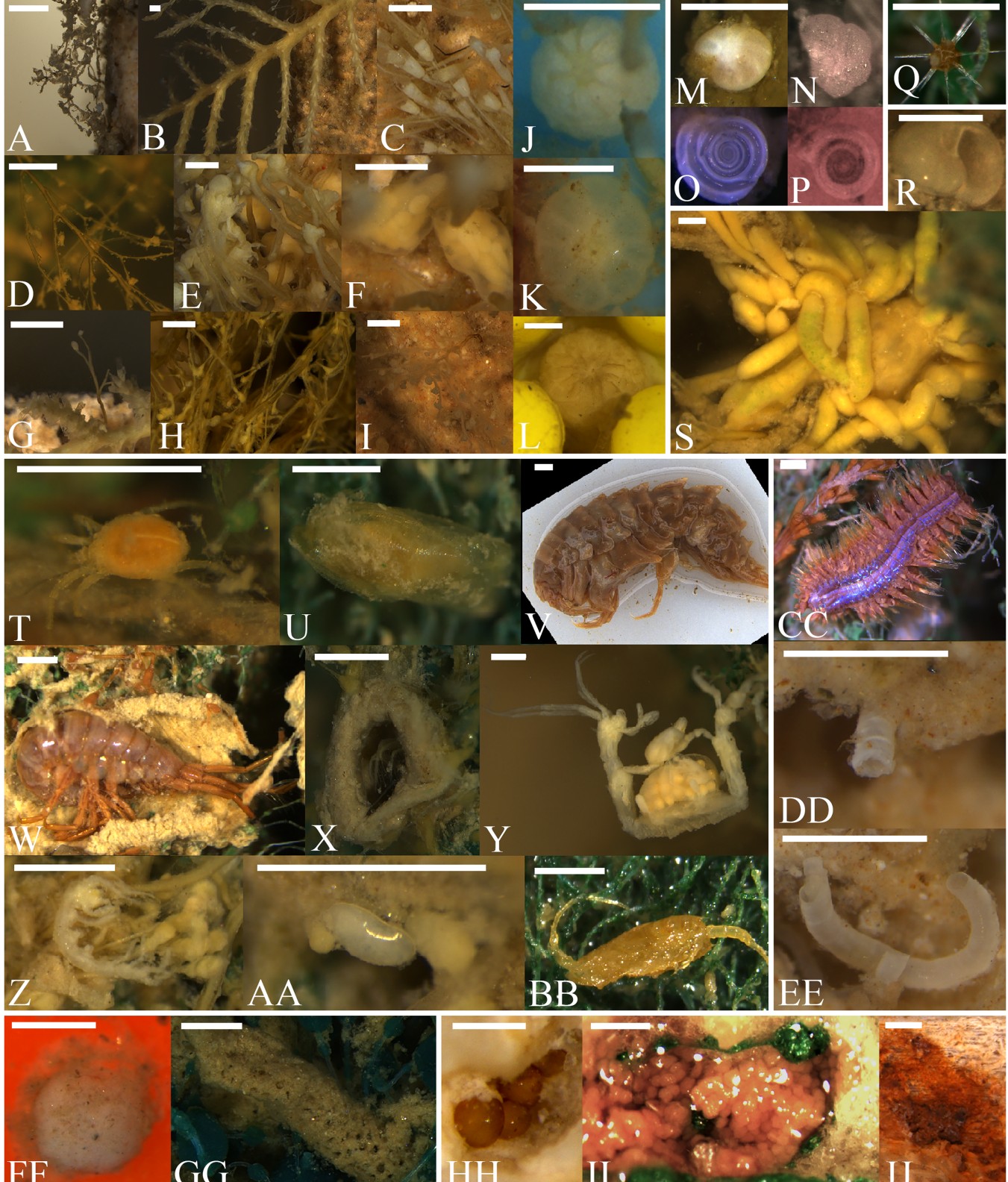

**Figure 3 Recruits found on all deployed substratum types in the Labrador Sea (Newfoundland and Labrador, Canada) arranged by phylum.** Scale bars (white) represent 1 mm for A-L, T-BB, CC-EE; 0.5 mm for M-P, Q, R-S, F-GG, HH-JJ. Phylum Cnidaria (A-L), including Hydrozoa (A-I) and Anthozoa (J-L): (A) Campanulariidae msp. 1, (B) Campanulariidae msp. 2, (C) Campanulariidae msp. 3, (D) Campanulariidae msp. 4,
**Figure 3 (continued)**

(E) *Eudendrium* msp. 1 colony, (F) *Eudendrium* msp. 1 gonozooids, (G) Epibiotic *Eudendrium* msp. 1 on Campanulariidae msp. 2 (see B), (H) *Eudendrium* msp. 2 colony, (I) Hydrozoa msp. 1 colony, (J) Octocorallia msp. 1 primary polyp, (K) Octocorallia msp. 1, (L) Actiniaria msp. 1. Phylum Foraminifera (M-P): (M) Foraminifera msp. 1, (N) Foraminifera msp. 2, (O) Foraminifera msp. 3, and (P) Foraminifera msp. 4. Scale bar (white) is applicable to M–P. Phylum Radiolaria: (Q) Radiolaria msp. 1. Phylum Mollusca (R-S): (R) Gastropoda msp. 1, and (S) Gastropoda msp. 2 egg masses on the base of a Campanulariidae msp. 2 colony (B). Phylum Arthropoda (T-BB): (T) Halacaridae msp. 1, mite within mesh substratum, (U) Ostracoda msp. 1, from between the mesh substratum sheets, (V) Gammaridea msp. 1, amphipod (W) Gammaridea msp. 2, tube-dwelling amphipod, after removal of tube on one side (X) Same morphospecies but different individual from W, without removal from tube, (Y) Caprellidae msp. 1, gravid female with brood pouch, (Z) Caprellidae msp. 1 cluster of smaller individuals, (AA) Isopoda msp. 1, and (BB) Copepoda msp. 1, found within mesh. Phylum Annelida (CC-EE): (CC) Polychaeta msp. 1, free living polychaete photographed after removal from mesh substratum sheets, (DD) Polychaeta msp. 2, tube-dwelling polychaete (tube only), and (EE) Same as DD, after removal from stone substratum. Phylum Porifera (FF-GG): (FF) Porifera msp. 1, and (GG) Porifera msp. 3. Morphospecies of unknown phylum (HH-JJ): (HH) Unknown msp. 1 cluster, subsurface on stone substratum type, (II) Unknown msp. 2, a possible egg mass on mesh substratum, and (JJ) Unknown msp. 3, a biological aggregate in a divot on wood substratum type.

**Table 1 Frequency of occurrence of all morphospecies found across the four substratum types deployed at four geographic sites.**

| Morphospecies | Substratum type | | | | Geographic site | | | |
|---|---|---|---|---|---|---|---|---|
| | *Mesh* | *Plastic* | *Stone* | *Wood* | *Site 1* | *Site 2* | *Site 3* | *Site 4* |
| Polychaeta msp. 1 | 1 | 0 | 1 | 0 | 0 | 0 | 1 | 1 |
| Polychaeta msp. 2 | 2 | 0 | 0 | 0 | 1 | 0 | 1 | 0 |
| Caprellidae msp. 1 | 2 | 1 | 0 | 0 | 1 | 0 | 0 | 2 |
| Copepoda msp. 1 | 4 | 3 | 0 | 1 | 1 | 2 | 2 | 3 |
| Gammaridea msp. 1 | 1 | 1 | 1 | 0 | 0 | 0 | 3 | 0 |
| Gammaridea msp. 2 | 1 | 0 | 0 | 0 | 1 | 0 | 0 | 0 |
| Halacaridae msp. 1 | 1 | 0 | 0 | 0 | 1 | 0 | 0 | 0 |
| Isopoda msp. 1 | 1 | 0 | 0 | 0 | 0 | 0 | 0 | 1 |
| Ostracoda msp. 1 | 1 | 0 | 0 | 0 | 0 | 0 | 0 | 1 |
| Actiniaria msp. 1 | 2 | 1 | 1 | 0 | 2 | 0 | 0 | 2 |
| Campanulariidae msp. 1 | 0 | 0 | 1 | 0 | 0 | 1 | 0 | 0 |
| Campanulariidae msp. 2A | 3 | 3 | 3 | 1 | 3 | 0 | 3 | 4 |
| Campanulariidae msp. 2B | 0 | 3 | 2 | 1 | 2 | 0 | 1 | 3 |
| Campanulariidae msp. 3 | 4 | 4 | 4 | 3 | 4 | 4 | 3 | 4 |
| Campanulariidae msp. 4 | 2 | 0 | 0 | 0 | 1 | 0 | 0 | 1 |
| *Eudendrium* msp. 1 | 3 | 3 | 3 | 1 | 3 | 0 | 3 | 4 |
| *Eudendrium* msp. 2 | 1 | 0 | 0 | 0 | 0 | 0 | 0 | 1 |
| Hydrozoa msp. 1 | 0 | 0 | 1 | 0 | 0 | 0 | 0 | 1 |
| Octocorallia msp. 1 | 2 | 2 | 2 | 0 | 3 | 0 | 0 | 3 |
| Foraminifera msp. 1 | 4 | 1 | 3 | 1 | 3 | 1 | 3 | 2 |
| Foraminifera msp. 2 | 1 | 0 | 1 | 0 | 0 | 0 | 0 | 2 |
| Foraminifera msp. 3 | 2 | 0 | 0 | 0 | 0 | 1 | 0 | 1 |
| Foraminifera msp. 4 | 1 | 1 | 0 | 0 | 0 | 0 | 2 | 0 |
| Gastropoda msp. 1 | 1 | 0 | 0 | 0 | 0 | 0 | 0 | 1 |
| Gastropoda msp. 2 | 1 | 0 | 0 | 0 | 0 | 0 | 0 | 1 |
| Porifera msp. 1 | 1 | 1 | 0 | 0 | 0 | 1 | 1 | 0 |

(Continued)

| Morphospecies | Substratum type | | | | Geographic site | | | |
|---|---|---|---|---|---|---|---|---|
| | *Mesh* | *Plastic* | *Stone* | *Wood* | *Site 1* | *Site 2* | *Site 3* | *Site 4* |
| Porifera msp. 2 | 2 | 0 | 0 | 0 | 1 | 1 | 0 | 0 |
| Porifera msp. 3 | 1 | 0 | 0 | 0 | 1 | 0 | 0 | 0 |
| Radiolaria msp. 1 | 3 | 1 | 2 | 0 | 0 | 2 | 1 | 3 |
| Unknown msp. 1 | 0 | 1 | 3 | 0 | 2 | 1 | 0 | 1 |
| Unknown msp. 2 | 1 | 0 | 0 | 0 | 0 | 0 | 0 | 1 |
| Unknown msp. 3 | 1 | 0 | 0 | 1 | 0 | 1 | 1 | 0 |

**Note:**
For substratum type, the number indicates at how many sites the morphospecies occurred on that substratum. For geographic site, the number indicates on how many substratum types the morphospecies occurred at that site. A zero indicates the morphospecies was not present.

However, the colonial hydrozoan Campanulariidae msp. 3 and Foraminifera msp. 1 colonized all substratum types when pooling sites, and all sites when pooling substratum types. Of all morphospecies, Campanulariidae msp. 3 dominated with 114,821 recorded individuals (Table 2; Table S4), a density of $4.2 \pm 1.4$ ind cm$^{-2}$ (Table S5) and the most surface and canopy coverage ($18 \pm 11\%$ and $12 \pm 9\%$ respectively). Four mspp (Halacaridae msp. 1, Ostracoda msp. 1, Gastropoda msp. 1, and Porifera msp. 3) occurred as a single individual or colony across all substrata and geographic sites.

Phylum Cnidaria exhibited the highest richness (Fig. 4) as well as highest abundance (130,859 ind; Table 2), density ($2.81 \pm 2.80$ ind cm$^{-2}$; Table S5), and surface and canopy coverage ($10.7 \pm 13.3\%$ and $10.1 \pm 14.8\%$ respectively; Fig. 5; Tables S6, S7, S8). While orders of magnitude fewer, foraminifers were the second most abundant phylum (262 ind), followed by arthropods (131 ind), molluscs (35 ind), radiolarians (8 ind), poriferans (5 ind), and annelids (4 ind; Table 2). Density of all these other phyla was below 0.01 ind cm$^{-2}$ (Table S5). Cover of the substrata for non-cnidarian phyla were $1.6 \pm 1.5\%$ at the base and $0.3 \pm 1.1\%$ at the canopy; (Tables S6, S7, S8), but consistently higher at the base than at the canopy (Fig. 5).

## Effect of substratum type and features

When analysing colonization by substratum type irrespective of geographic site, plastic was the most colonized material with a total of 41,763 individuals (morphospecies pooled; Table 2), and a density of $50.5 \pm 28.8$ ind cm$^{-2}$ (Table S5). It was followed by stone (41,549 ind, $31.7 \pm 27.7$ ind cm$^{-2}$), mesh (41,260 ind, $26.0 \pm 31.1$ ind cm$^{-2}$), and wood (3,152 ind, $4.8 \pm 6.2$ ind cm$^{-2}$). Total abundance differed significantly among substrata ($p < 0.001$; Table S9), as did density ($p < 0.001$). Plastic exhibited the tightest clustering in total abundance (Fig. 6). Total abundance significantly differed between mesh *vs.* plastic and mesh *vs.* stone, as well as between plastic *vs.* wood ($p < 0.05$ for all, Table S9). Density was also significantly higher on mesh than on plastic or stone as well as higher on plastic than on wood (Table S9). Plastic also exhibited the highest abundance and density of any one morphospecies, *i.e.*, Campanulariidae msp. 3 (37,347 ind, $7.6 \pm 4.3$ ind cm$^{-2}$; Tables S4, S5).
**Table 2 Morphospecies abundance examined as total number of individuals or colonies.** Bolded lines of numbers represent the sum of individuals or colonies in each phylum.

| Morphospecies | Global | Substratum type | | | | Geographic site | | | |
|---|---|---|---|---|---|---|---|---|---|
| | Total | Mesh | Plastic | Stone | Wood | Site 1 | Site 2 | Site 3 | Site 4 |
| **Annelida** | **4** | **2** | **0** | **1** | **0** | **0** | **0** | **2** | **1** |
| Polychaeta msp. 1 | 2 | 1 | 0 | 1 | 0 | 0 | 0 | 1 | 1 |
| Polychaeta msp. 2 | 2 | 1 | 0 | 0 | 0 | 0 | 0 | 1 | 0 |
| **Arthropoda** | **131** | **98** | **15** | **13** | **1** | **22** | **21** | **31** | **53** |
| Caprellidae msp. 1 | 21 | 20 | 1 | 0 | 0 | 1 | 0 | 0 | 20 |
| Copepoda msp. 1 | 67 | 58 | 4 | 0 | 1 | 16 | 21 | 3 | 23 |
| Gammaridea msp. 1 | 28 | 5 | 10 | 13 | 0 | 0 | 0 | 28 | 0 |
| Gammaridea msp. 2 | 4 | 4 | 0 | 0 | 0 | 4 | 0 | 0 | 0 |
| Halacaridae msp. 1 | 1 | 1 | 0 | 0 | 0 | 1 | 0 | 0 | 0 |
| Isopoda msp. 1 | 9 | 9 | 0 | 0 | 0 | 0 | 0 | 0 | 9 |
| Ostracoda msp. 1 | 1 | 1 | 0 | 0 | 0 | 0 | 0 | 0 | 1 |
| **Cnidaria** | **130,859** | **40,875** | **41,745** | **41,422** | **3,149** | **56,841** | **12,342** | **8,728** | **49,281** |
| Actiniaria msp. 1 | 5 | 3 | 1 | 0 | 0 | 1 | 0 | 0 | 3 |
| Campanulariidae msp. 1 | 16 | 0 | 0 | 13 | 0 | 0 | 13 | 0 | 0 |
| Campanulariidae msp. 2A | 553 | 168 | 186 | 113 | 73 | 160 | 0 | 16 | 364 |
| Campanulariidae msp. 2B | 3,913 | 0 | 1,840 | 1,224 | 850 | 951 | 0 | 139 | 2,822 |
| Campanulariidae msp. 3 | 114,821 | 33,721 | 37,347 | 38,145 | 2,135 | 54,563 | 12,329 | 6,922 | 37,533 |
| Campanulariidae msp. 4 | 17 | 16 | 0 | 0 | 0 | 1 | 0 | 0 | 15 |
| *Eudendrium* msp. 1 | 7,123 | 2,567 | 2,368 | 1,922 | 91 | 1,160 | 0 | 1,650 | 4,138 |
| *Eudendrium* msp. 2 | 4,396 | 4,396 | 0 | 0 | 0 | 0 | 0 | 0 | 4,396 |
| Hydrozoa msp. 1 | 2 | 0 | 0 | 2 | 0 | 0 | 0 | 0 | 2 |
| Octocorallia msp. 1 | 12 | 4 | 3 | 4 | 0 | 4 | 0 | 0 | 7 |
| **Foraminifera** | **262** | **238** | **2** | **8** | **1** | **137** | **6** | **91** | **15** |
| Foraminifera msp. 1 | 255 | 233 | 1 | 7 | 1 | 137 | 5 | 89 | 11 |
| Foraminifera msp. 2 | 2 | 1 | 0 | 1 | 0 | 0 | 0 | 0 | 2 |
| Foraminifera msp. 3 | 3 | 3 | 0 | 0 | 0 | 0 | 1 | 0 | 2 |
| Foraminifera msp. 4 | 2 | 1 | 1 | 0 | 0 | 0 | 0 | 2 | 0 |
| **Mollusca** | **35** | **35** | **0** | **0** | **0** | **0** | **0** | **0** | **35** |
| Gastropoda msp. 1 | 1 | 1 | 0 | 0 | 0 | 0 | 0 | 0 | 1 |
| Gastropoda msp. 2 | 34 | 34 | 0 | 0 | 0 | 0 | 0 | 0 | 34 |
| **Porifera** | **5** | **4** | **0** | **0** | **0** | **2** | **1** | **1** | **0** |
| Porifera msp. 1 | 2 | 1 | 0 | 0 | 0 | 0 | 0 | 1 | 0 |
| Porifera msp. 2 | 2 | 2 | 0 | 0 | 0 | 1 | 1 | 0 | 0 |
| Porifera msp. 3 | 1 | 1 | 0 | 0 | 0 | 1 | 0 | 0 | 0 |
| **Radiolaria** | **8** | **5** | **1** | **2** | **0** | **0** | **2** | **2** | **4** |
| Radiolaria msp. 1 | 8 | 5 | 1 | 2 | 0 | 0 | 2 | 2 | 4 |
| **Unknown** | **108** | **3** | **0** | **103** | **1** | **99** | **3** | **1** | **4** |
| Unknown msp. 1 | 104 | 0 | 0 | 103 | 0 | 99 | 2 | 0 | 2 |
| Unknown msp. 2 | 2 | 2 | 0 | 0 | 0 | 0 | 0 | 0 | 2 |
| Unknown msp. 3 | 2 | 1 | 0 | 0 | 1 | 0 | 1 | 1 | 0 |

**Note:**

Zero indicates the morphospecies was not present. Errors are standard deviation (if absent, morphospecies occurred only once).

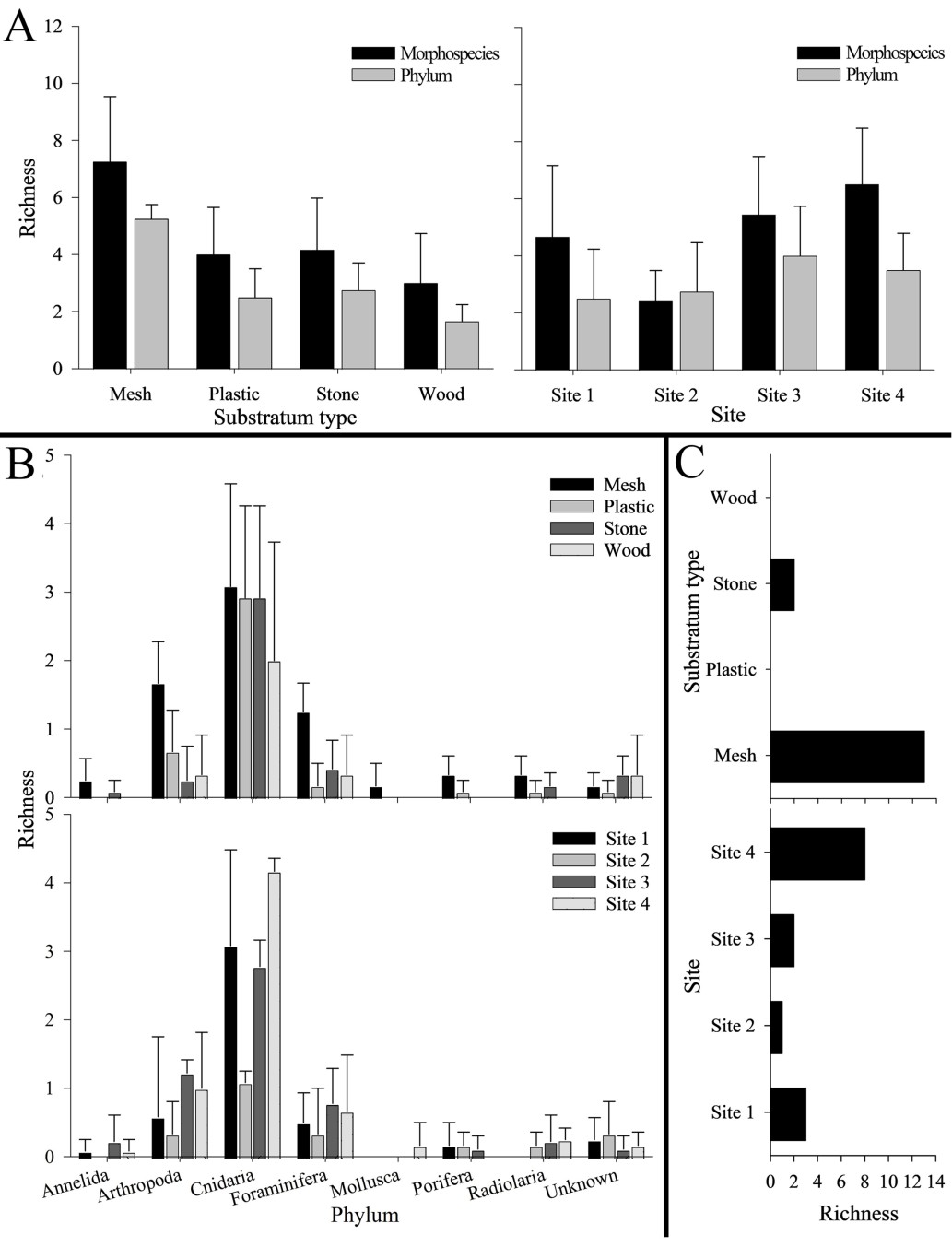

**Figure 4  Number of morphospecies, phyla, and unique morphospecies (*i.e.*, richness) across the four settlement frame substratum types (see Fig. 1) and four deployment sites (see Fig. 2) in Labrador Sea (Newfoundland and Labrador, Canada).** No bar indicates none present. Error bars represent standard deviation. (A) Morphospecies and phylum richness. Left: by substratum type. Right: by site. (B) Morphospecies richness within each phylum. Top: by substratum type. Bottom: by site. (C) Unique morphospecies richness. Top: by substratum type. Bottom: by site.

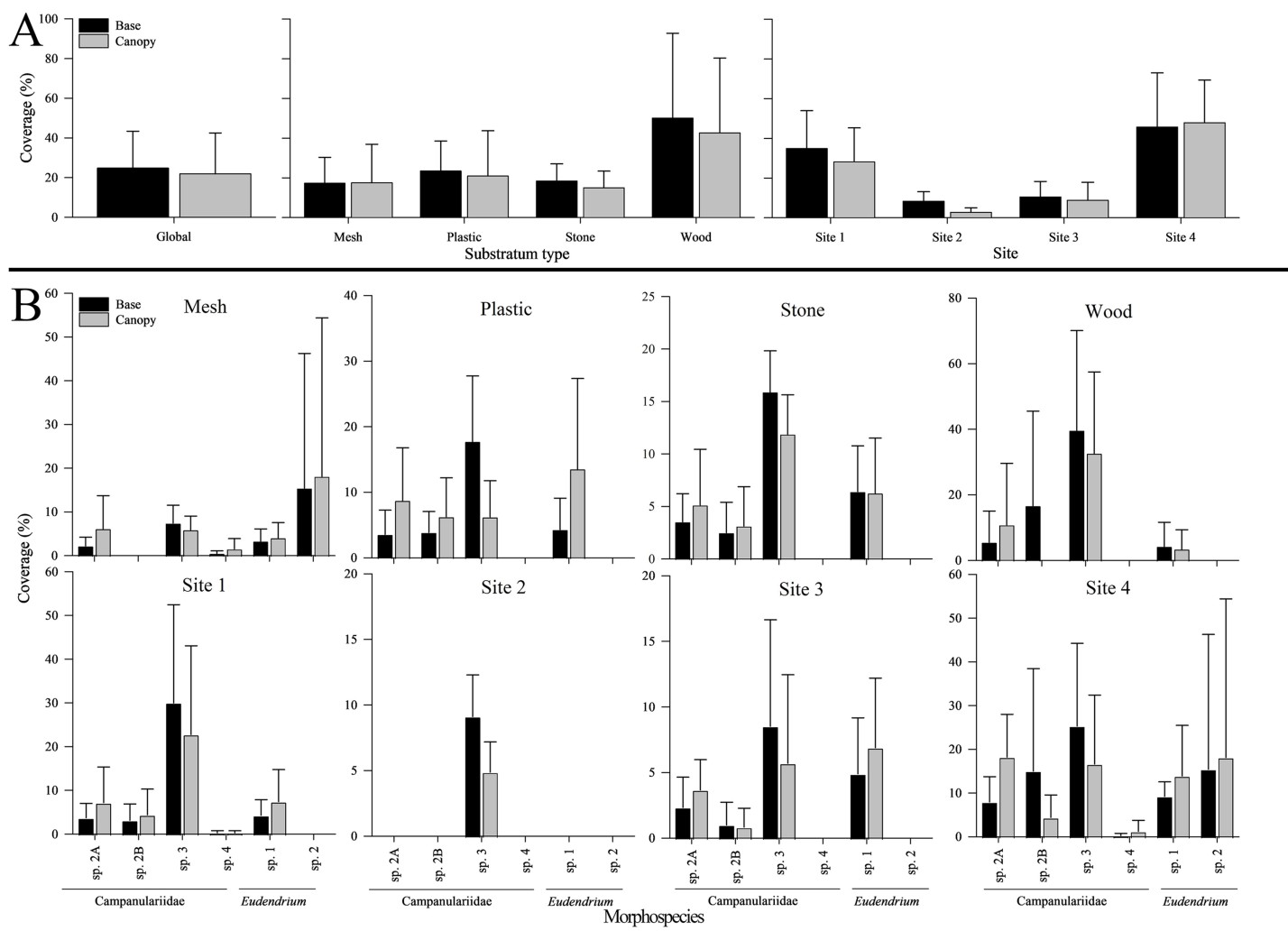

**Figure 5 Base and canopy cover exhibited by five common hydrozoans, examined by settlement frame substratum types (mesh, plastic, stone, wood) and by deployment site (Sites 1, 2, 3, 4) in the Labrador Sea (Newfoundland and Labrador, Canada).** Error bars indicate standard deviation. Note the differing scales of y-axes.               

Surface coverage differed significantly among substrata at both the base and canopy ($p < 0.001$ for both; Table S9). Wood was the most covered when combining all morphospecies, both at the base ($50.3 \pm 42.6\%$) and canopy ($42.8 \pm 37.5\%$), followed by plastic ($23.6 \pm 14.9\%$ and $21.1 \pm 22.6\%$; Fig. 5). Stone was covered more than mesh at the base ($18.6 \pm 8.4\%$ and $17.5 \pm 12.9\%$, respectively), whereas the inverse occurred at the canopy ($15.2 \pm 8.2\%$ and $17.7 \pm 19.2\%$, respectively) (Fig. 5).

Overall, mesh supported the highest total richness, with 26 mspp spanning seven phyla (and two unknown mspp). Plastic and stone were each colonized by 13 mspp from five phyla (and one unknown msp), with six mspp from three phyla colonizing wood (as well as one unknown msp). Mesh also harbored the highest morphospecies and phylum richness per substratum block ($7.3 \pm 2.3$ mspp and $5.3 \pm 0.5$ phyla block$^{-1}$), followed by stone, plastic, and wood (Fig. 4). Morphospecies richness differed significantly among

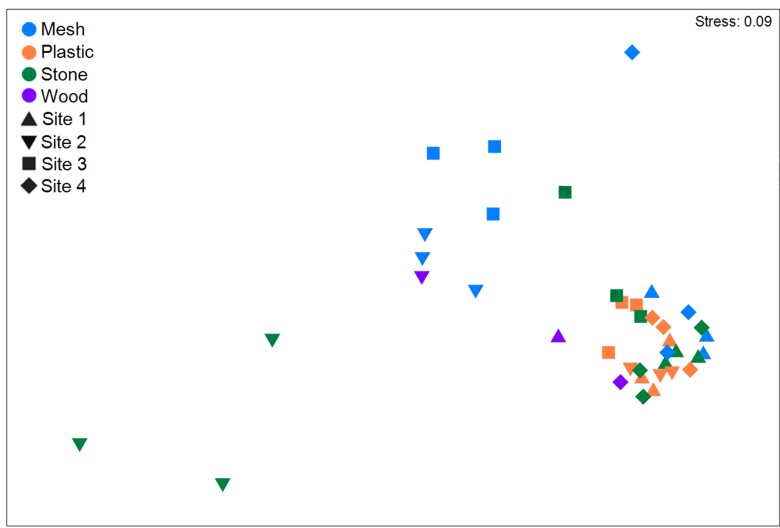

**Figure 6** Non-metric multidimensional scaling (nMDS) using Bray-Curtis similarity coefficients of total abundance of recruits of all morphospecies to the four substratum types present at four geographic sites in the Labrador Sea (Canada).

substratum types, as did phylum richness ($p < 0.001$ for both; Table S9). Four ubiquitous mspp occurred on all substratum types: Campanulariidae msp. 2 and msp. 3, *Eudendrium* msp. 1, and Foraminifera msp. 1 (*i.e.*, 13% of all mspp combined; Table 1). Conversely, 15 mspp occurred on only one substratum (*i.e.*, 48% of all mspp), chiefly on mesh ($n = 13$; 42%), and stone ($n = 2$; 6%), whereas no morphospecies occurred exclusively on plastic or wood (Fig. 4C).

### Sessile morphospecies spatial recruitment and colonization patterns

Twenty mspp colonized surface locations and microhabitats (Fig. 7). Eleven mspp colonized just one surficial location per substratum type; of these, four mspp colonized just one location globally (described below; Table 3). Eleven mspp occurred in just one microhabitat per substratum type; six mspp colonized just one microhabitat globally (four singletons, and two which occurred multiple times in the same microhabitat). Overall, 16 mspp colonized the centre location, irrespective of substratum type and site (~89%) and 12 mspp colonized edges (~60%). Sixteen mspp colonized the unsheltered microhabitat irrespective of substratum type and site (~80%), whereas 18 colonized the sheltered microhabitats (90%; Fig. 7; Table 3). Only one msp, Unknown msp. 3, bored into the wood substratum (*i.e.*, sheltered microhabitat).

Three colonial hydrozoans (Campanulariidae msp. 3, msp. 2, and *Eudendrium* msp. 1) colonized almost all locations and microhabitats available on the substrata (Fig. 7A). The most abundant and opportunistic colonizer (Campanulariidae msp. 3) did not display location preferences, and colonies extended across unsheltered and sheltered microhabitats, except for sheltered microhabitats on stone at Site 2, mesh at Site 1, and wood. Campanulariidae msp. 2 also showed little location preference, except for mesh edges at Site 3. Biota colonized almost all microhabitats, except for sheltered microhabitats

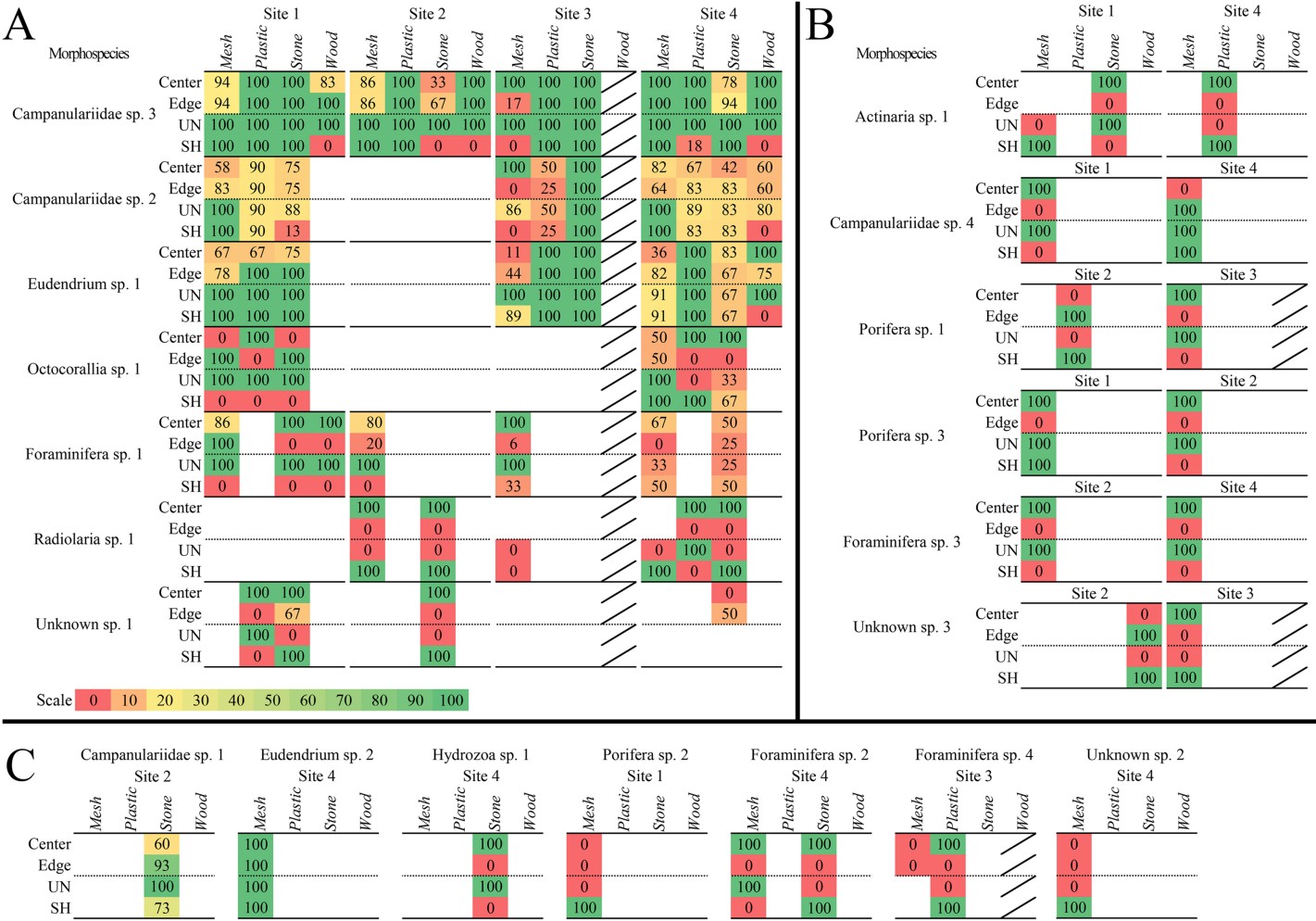

**Figure 7 Heat maps of recruitment locations and microhabitats of all sessile morphospecies.** Location and microhabitat recruitment is expressed as the percentage of block faces on which each morphospecies occurred in that location or microhabitat out of the total number of block faces on which the morphospecies occurred overall. Morphospecies that occurred in multiple locations or microhabitats on each block face had each occurrence counted, so all locations and microhabitats are accounted for. (A) Morphospecies that recruited broadly to surface locations and microhabitats, on three or more substratum types or sites. (B) Morphospecies that recruited more narrowly to locations and microhabitats at two sites. (C) Morphospecies that recruited to one location or microhabitat and/or at one site. Diagonal bar indicates substratum types that were not analyzed.

on stone at Site 3 and wood. *Eudendrium* msp. 1 colonized corner, edge, and centre locations with no exceptions; all microhabitats available were colonized except the sheltered microhabitat on wood. For a detailed description of location and microhabitat utilization, refer to Supplementary results "Surface location and microhabitat colonization pattern detail".

### *Motile morphospecies spatial recruitment and colonization patterns*

Ten motile mspp utilized specific surface locations and microhabitats. Seven mspp occurred in the centre location, irrespective of substratum type and site (~78%), four on edges (~44%), and two in corners (~22%; Table 3). Five mspp occurred in unsheltered

**Table 3 Recruitment location and microhabitat preferences of motile morphospecies.**

| Morphospecies | Site | Substratum type | Location (%) | | Microhabitat (%) | |
|---|---|---|---|---|---|---|
| | | | *Centre* | *Edge* | *Unsheltered* | *Sheltered* |
| Polychaeta msp. 2 | 1 | Mesh | 100 | 0 | 0 | 100 |
| Polychaeta msp. 2 | 3 | Mesh | 0 | 0 | 0 | 100 |
| Polychaeta msp. 2 | 4 | Stone | 100 | 0 | 0 | 100 |
| Caprellidae msp. 1 | 4 | Mesh | 0 | 50 | 50 | 50 |
| Copepoda msp. 1 | 1 | Mesh | 80 | 20 | 0 | 100 |
| Copepoda msp. 1 | 2 | Mesh | 50 | 36 | 50 | 100 |
| Copepoda msp. 1 | 2 | Plastic | 100 | 0 | 0 | 100 |
| Copepoda msp. 1 | 3 | Mesh | 0 | 0 | 60 | 100 |
| Copepoda msp. 1 | 3 | Plastic | 100 | 0 | 0 | 100 |
| Copepoda msp. 1 | 4 | Mesh | 80 | 20 | 0 | 100 |
| Copepoda msp. 1 | 4 | Plastic | 100 | 0 | 0 | 100 |
| Copepoda msp. 1 | 4 | Wood | 100 | 0 | 100 | 0 |
| Gammaridea msp. 1 | 3 | Mesh | 60 | 20 | 60 | 33 |
| Gammaridea msp. 1 | 3 | Plastic | 100 | 0 | 13 | 88 |
| Gammaridea msp. 1 | 3 | Stone | 100 | 0 | 100 | 0 |
| Gammaridea msp. 2 | 1 | Mesh | 33 | 67 | 100 | 67 |
| Halacaridae msp. 1 | 1 | Mesh | 0 | 100 | 0 | 100 |
| Isopoda msp. 1 | 4 | Mesh | 100 | 0 | 100 | 0 |
| Ostracoda msp. 1 | 4 | Mesh | 100 | 0 | 0 | 100 |
| Gastropoda msp. 1 | 4 | Mesh | 100 | 0 | 0 | 100 |
| Gastropoda msp. 3 | 4 | Mesh | 0 | 0 | 0 | 100 |

**Note:**
Measurements are expressed as the percentage of block faces on which each morphospecies occurred in that location or microhabitat out of the total number of block faces on which the morphospecies occurred overall. Morphospecies that occurred in multiple locations or microhabitats on each block face had each occurrence counted, so all locations and microhabitats are accounted for.

microhabitats irrespective of substratum type or site (~56%), and nine mspp in sheltered microhabitats (90%) (see details Table 3).

## Effect of geographic site and associated factors

Settlement frames were positioned at differing depths and altitudes across sites (Fig. 2). When combining all substrata within each geographic site, Site 1 had the highest abundance of recruits/colonizers (57,101 ind; Table 2; Table S4) and highest density (52.1 ± 35.8 ind cm$^{-2}$; Table S5), followed by Site 4, Site 2, and Site 3. Total abundance differed significantly among sites, as did density ($p < 0.001$ for both; Table S9). In nMDS, Sites 1 and 4 clustered together, while Sites 2 and 3 clustered more loosely into two clusters (Fig. 6). Site 1 differed significantly from Site 2 and Site 3; Site 2 differed significantly from Site 3 and Site 4; and Site 3 differed significantly from Site 4 (summarized in Table S9).

Total richness across substrata was highest at Site 4 (22 mspp from 6 phyla), followed by Site 1 (16 mspp from five phyla), Site 3 (13 mspp from 6 phyla), and Site 2 (10 mspp from five phyla), whereas in morphospecies richness per substratum block, Site 4 was followed

by Site 3, Site 1, and Site 2 (Fig. 4). Morphospecies richness differed significantly among sites ($p < 0.001$; Table S9). Shannon diversity was highest at Site 3 (H' = 0.74 ± 0.47), followed by Site 4 (H' = 0.53 ± 0.27), Site 2 (H' = 0.32 ± 0.31), and Site 1 (H' = 0.18 ± 0.13). Shannon diversity differed significantly among sites ($p < 0.001$), with Site 1 differing significantly from Site 2 and Site 3, and Site 2 differing significantly from Site 3 and Site 4 (Table S9).

The four geographic sites had three mspp in common: Campanulariidae msp. 3, Copepoda msp. 1, and Foraminifera msp. 1 (*i.e.*, 10% of all mspp; Table 1). Conversely, 14 mspp (*i.e.*, 45% of all mspp) occurred at only one site, with the highest number of unique morphospecies at Site 4 ($n = 8$, 25% of all mspp, 505 m) and the fewest at Site 2 ($n = 1$, 3% of all mspp, 960 m; Fig. 4). Highest total abundance of Campanulariidae msp. 3 (54,563 ind) and density (9.3 ± 4.7 ind cm$^{-2}$) occurred at Site 1, followed by Site 4, Site 2, and finally Site 3 (Table 2; Tables S4, S5). Abundance and density of Campanulariidae msp. 3 was significantly different between sites ($p < 0.001$). More broadly, only arthropods, cnidarians, and foraminifers were recorded at all sites (Fig. 4).

### Growth patterns and lifestyles

Colonial forms dominated at the substratum level, averaging between 1.7 ± 1.2 and 3.3 ± 1.3 per block, with correspondingly lower averages for unitary morphospecies between 0.7 ± 0.6 and 3.8 ± 1.0 per block. When examined by site, colonial morphospecies consistently dominated as well, averaging between 1.3 ± 0.2 and 3.5 ± 0.6 per block, in contrast to a range of 0.8 ± 1.2 to 2.4 ± 1.4 per block for unitary morphospecies.

Sessile morphospecies dominated at the substratum level, averaging between 2.3 ± 1.5 and 5.2 ± 1.5 per block whereas correspondingly lower averages between 0.3 ± 0.6 and 1.9 ± 0.7 per block characterized motile morphospecies. When examined by site, sessile morphospecies dominated as well, averaging from 1.8 ± 0.9 to 5.1 ± 1.1 per block whereas motile morphospecies ranged from 0.3 ± 0.5 to 1.3 ± 1.0 per block.

### Reproductive status

A few individuals harboured oocytes and embryos. A mature female Caprellidae msp. 1 with a brood pouch carried visible embryos (Fig. 3E). Individuals of two mspp of colonial hydrozoans, Campanulariidae msp. 3 and *Eudendrium* msp. 1, had mature gonozooids containing visible oocytes (Fig. 3A).

### Epibiosis

There were 16 occurrences of epibiosis, all on three colonial hydrozoan hosts (Campanulariidae msp. 2, *Eudendrium* mspp. 1 and 2). The associations occurred primarily at Site 4 ($n = 12$), with some at Sites 1 and 3 ($n = 2$ each). One gastropod mollusc that occurred only as egg masses, Gastropoda msp. 2, occurred exclusively as an epibiont of Campanulariidae msp. 2. Five mspp occurred occasionally as epibionts: the benthic Foraminifera msp. 1, two colonial hydrozoans (Campanulariidae msp. 3 and *Eudendrium* msp. 1), the Caprellidae msp. 1, and individual eggs of Unknown msp. 1. The most common epibiont, Foraminifera msp. 1, was observed twice each on Campanulariidae

msp. 2, *Eudendrium* mspp. 1 and 2. *Eudendrium* msp. 1 also occurred commonly as an epibiont on Campanulariidae msp. 2 (n = 4). Similarly, Campanulariidae msp. 2 also occurred once as an epibiont on *Eudendrium* msp. 1. Individual eggs (Unknown msp. 1) occurred once on Campanulariidae msp. 2. Of the motile morphospecies, Caprellidae msp. 1 was observed on *Eudendrium* msp. 1 three times.

## DISCUSSION

### Trends in abundance, richness, diversity, and coverage

At the temporal scales studied (about 1 year), colonial hydrozoans emerged as the dominant recruits on all substrata and at all sites/depths, contributing almost the entirety of overall abundance and cover of substratum surfaces and canopies. Numerous studies document colonial hydrozoans as pioneer recruits to artificial settlement frames and anthropogenic structures in tropical, temperate, and polar shallow waters (*Boero, 1984*; *Ronowicz, 2007*; *Ronowicz, Wlodarska-Kowalczuk & Kukliński, 2008*, *2013*; *Calder et al., 2021*). Arctic deep-sea taxa apparently follow this pattern, based on the present findings and those of *Meyer-Kaiser et al. (2019)*, who reported a colonial hydrozoan (*Halisiphonia arctica*) among the earliest recruits to brick and plastic substrata at 2,500 m depth in the Fram Strait (west of Svalbard, Norway). Hydrozoans were also the only sessile morphospecies found here to harbor eggs and to host epibionts, attesting to their rapid development and sexual maturation. Similarly, *Meyer-Kaiser et al. (2022)* observed three reproductively mature species of hydrozoans on settlement plates deployed over a comparable time span in the Fram Strait. Considering the near absence of these same hydrozoan morphospecies on well-established hard bottoms in nearby habitats (*Wolvin, Hamel & Mercier, 2025*), these erect hydrozoan colonies may act as a crucial first stage of succession in the establishment of hard bottom deep-sea benthic communities. Other taxa may rely on their presence for recruitment away from the boundary layer (*e.g.*, caprellids and foraminifers) and some might graze them (*e.g.*, gastropods) and eventually overtake them in the long term. Gastropod nudibranchs prey on hydrozoans (*Martin, 2003*) and they occur at bathyal depths in the region (*Penney, Hamel & Mercier, 2020*), which could suggest egg masses were laid by feeding adults on associated species of hydrozoans, similar to corallivorous nudibranchs laying eggs on deep-sea octocorals in the NW Mediterranean (*Priori et al., 2015*). Similarly, *Lutze & Thiel (1989)* reported that some foraminifer species preferentially position themselves on elevated substrata for better access to food, and caprellids use or even preferentially select biogenic structures such as hydrozoans as substrata in shallow North Atlantic studies (*McCain, 1968*; *Caine, 1998*). Such preferences in epibiotic relationships could define the first steps of succession in such an early community and suggests that they can create temporary localized diversity hotspots.

### Effect of substratum type and features

Differences in morphospecies and phylum richness were observed, as well as density between substrata, suggesting that features such as the surface material, locations, and microhabitats available to larvae play a role in recruitment and colonization patterns. Higher richness characterized the complex three-dimensional structure of the mesh

substratum at both the morphospecies and phylum levels. Mesh also hosted the most unique morphospecies, although it did not dominate in recruit abundance, density, or surface coverage. The comparatively loose and flexible structure of the mesh, with its folded surface, may accommodate more recruits but could impede their growth and expansion. Accordingly, *Girard, Lacharité & Metaxas (2016)* reported that no single taxon dominated on complex substrata, and that lower hydrozoan biomass (*i.e.*, abundance and coverage in the present study) occurred on complex substrata in their 4-y study in the Gulf of Maine (USA) at depths between 600–900 m. *Burkett et al. (2016)* deployed a recruitment experiment containing three substratum types at 595–777 m depths in Hydrate Canyon (Oregon, USA), and reported that epibenthic foraminifers never settled on wood, whereas polypropylene (comparable to plastic in the present study) hosted only 3% of total foraminifer abundance relative to mesh (97%). Here, foraminifers colonized other substrata, but only mesh harbored all four morphospecies, underscoring a general attraction to this more complex and pliable substratum.

Plastic supported the highest abundance and density of morphospecies per surface area, likely due to the three-dimensional, sheltered microhabitat created by projections on this substratum. Similarly, *Lacharité & Metaxas (2013)* reported greater recruitment of corals (*P. resedaeformis* and *P. arborea*) to the structurally complex plastic components of the collector frame. The presence of three-dimensional components (comparable to sheltered microhabitat) could result in minute changes in flow, food availability, and protection against predators. Most other recruitment experiments listing plastic as a substratum type used flat panels (without projections), which yielded results opposite to the present study. For instance, *Meyer-Kaiser et al. (2019)* reported higher abundance of recruits on brick panels (~stone in the present study) than on plastic panels deployed for 19 years, attributing it to more complex microhabitats on the former than the latter. This interpretation supports the suggestion that high recruit abundance on the plastic is due to microhabitat complexity rather than material; however, it has been shown as well that two species of *Bugula* bryozoan appear to exhibit a preference for plastic as a settlement substratum (*Pinochet, Urbina & Lagos, 2020*).

Wood had the highest surface coverage by all morphospecies combined, at both the base and canopy levels. Its smooth surface may have allowed expansion of recruits more easily than the more complex three-dimensional structures. Moreover, fine-scale flow around the frame could affect recruitment (*Mullineaux & Garland, 1993*; *Lacharité & Metaxas, 2013*), so that unsheltered and larger wood panels could improve the vertical and horizontal expansion by some taxa, mostly hydrozoans. The greater abundance of a colonial hydrozoan occurring both as erect, branching and horizontally-stolonizing colonies on wood here suggests that it could favor colonization success. The latter morphospecies could be similar to and as common as the deep-water Arctic hydrozoan *Stegopoma plicatile*, which densely covered settlement plates made of acrylic panels (a flat surface without sheltering features similar to the wood substratum used here) at 215 m in Kongsfjorden, western Spitsbergen, Norway (*Meyer et al., 2017*). Wood otherwise harbored just two morphospecies not among the four generalists, including the motile Copepoda msp. 1, which aligns with a recruitment experiment at the Lucky Strike

Hydrothermal Field (near Azores, Portugal) reporting a surface-dwelling copepod exclusive to wood (*Cuvelier et al., 2014*). However, contrary to that study, no wood-boring bivalves were found here. Wood specialist taxa may have been absent since frames were deployed at a latitude above the tree line, presumably offering a very limited supply of natural wood to the marine environment.

Stone did not dominate in any aspect of colonization, but it was one of two substrata that attracted unique morphospecies, including Campanulariidae msp. 1 and Hydrozoa msp. 1, suggesting a settlement preference. This aligns with *Ronowicz, Włodarska-Kowalczuk & Kukliński (2013)* who reported that shallow Arctic species of stolonate and erect hydrozoans (*Bougainvillia* cf. *superciliaris* and *Sarsia* sp.) occurred more commonly on rocks. Potentially, stone used in the present study (flat surface with minute pores) attracted flexible suspension feeders able to withstand the stronger flow associated with unsheltered substrata. Notably, Hydrozoa msp. 1 anchored in a sheltered microhabitat (pore) but grew vertically into the water column, perhaps exhibiting similar larval settlement selection in a sheltered location and then growing away from it as seen in some bryozoans (*Walters & Wethey, 1986*). Other recruits such as foraminifers, primary polyps of octocorals, and small actinarians also took advantage of the stone pores. Accordingly, *Girard, Lacharité & Metaxas (2016)* reported that actiniarians occurred only on simple substrata (comparable to stone) and not on complex mesh, though they did not test any other substratum types. Given that here actiniarians were found on both stone and plastic but in sheltered areas, these microhabitats may offer similarly attractive characteristics (*e.g.* flat surface with micro-shelter or anchorage points). However, four morphospecies apparently showed no substratum preference, including three colonial hydrozoans, two of which were campanulariids (Campanulariidae msp. 2A, 2B and 3), known to display little substratum preference during colonization (*Cornelius, 1982*). A third substratum-generalist hydrozoan in the present study (*Eudendrium* msp. 1) contrasted with *Ronowicz, Wlodarska-Kowalczuk & Kukliński (2008)*, who reported eudendriids on just one substratum type per species. *Wasserthal & Wasserthal (1973)* indicated that eudendriids reproduce using slime ropes along which planulae travel, increasing gregariousness with the parent colony, and they grow through horizontal stolonization (*Schuchert, 2008*). These characteristics could have driven expansion of substratum-specific recruits onto less preferred substrata within the same frame. Because the hydrozoan colonies had developed beyond the first recruit by the time of frame retrieval, the planulae potentially settled preferentially, but subsequent horizontal growth obscured these results.

Foraminifera msp. 1, the fourth substratum-generalist morphospecies recorded here, is morphologically similar to *Cibicidoides wuellersdorfi*, a common opportunistic epibenthic foraminifer, which *Meyer-Kaiser et al. (2019)* reported as a dominant recruit on both brick and plastic panels deployed in the deep sea of the Fram Strait. Earlier studies had characterized *C. wuellersdorfi* as a substratum generalist, colonizing hydroids, stones, tube worms, sponge skeletons, crinoids (*Lutze & Thiel, 1989*; *Linke & Lutze, 1993*) as well as mesh and plastic substrata at Hydrate Ridge, Oregon, USA (*Burkett et al., 2016*). Foraminifera msp. 1 commonly colonized centre locations and unsheltered areas of all substratum types, suggesting a preference for elevated or relatively exposed surfaces of any

material. This interpretation aligns with *Veillette et al. (2007)* reporting that most suspension-feeding foraminifers colonizing polymetallic nodules at ~5,000 m depth in the Clarion-Clipperton Fracture Zone (central Pacific) preferred raised over depressed microhabitats (~unsheltered and sheltered, respectively). Potentially, the larvae of these taxa do not have strict recruitment preferences, allowing them to colonize more substrata.

Most other morphospecies occupied block centres rather than edges, regardless of substratum material, including primary coral polyps, actiniarians, radiolarians, foraminifers, and poriferans. This supports the common exclusion of edges from settlement studies because of potential erosion and edge effects (*Bowden, 2005*; *Barnes, 2017*). Flows at block edges may be too strong or unpredictable for many taxa at settlement, or fauna at the edges may be eroded over time, since degradation was observed on the edges of the stones in the present study. However, some morphospecies occurred exclusively (*e.g.*, Campanulariidae msp. 4 on mesh at Site 4) or more commonly (*e.g.*, *Eudendrium* msp. 1 on plastic and stone at Site 1) on edge locations of some substratum types, which suggests that excluding edges could underestimate the richness, diversity, or abundance measured. More resilient or flexible suspension feeders, such as hydrozoans, may benefit from flow dynamics that decrease competition for the resources, giving them access to more adequate locations to colonize.

Conversely, more varied colonization preferences were observed between sheltered and unsheltered microhabitats. Actiniarians, radiolarians, and all three unknown morphospecies colonized sheltered microhabitats, whereas primary polyps of octocorals commonly occurred in unsheltered microhabitats, as did some hydrozoans. In contrast, *Lacharité & Metaxas (2013)* reported higher abundances of the coral *P. resedaeformis* in complex (~sheltered) habitat rather than on flat (~unsheltered) surfaces. This difference may reflect geographic or environmental differences between study areas. Importantly, it is not possible to fully separate the substratum materials, locations, and microhabitats from each other because their complex interplay affects fine-scale larval recruitment patterns, and as with the hydrozoans, it is possible that all recruits expanded beyond initial settlement location and obscured the results. In addition, a seeming preference for these features could be more a function of the benefits they convey towards successful recruitment through post-settlement processes than a true preference.

### Effect of geographic site and associated factors

Inter-site differences in abundance, density, surface coverage (at base and canopy), and morphospecies richness suggest that depth, geographic site, year, and frame altitude above bottom may act either independently or in combination. Phylum richness, a notable exception, did not differ across sites. Recruitment experiments in the deep sea have reported variation among sites that could reflect contributions of factors such as depth, frame altitude, and local water-mass characteristics (*Romano et al., 2014*; *Meyer-Kaiser et al., 2019*, *2022*). In Antarctic shallow-water recruitment experiments that included several sites, *Bowden et al. (2006)* also reported that variability in local conditions could influence recruitment and colonization; however, some variability was seasonal due to the shallow depth of the plate deployments. The present study, occurring in the deep sea, did

not measure local environmental conditions such as temperature and currents, which could also have contributed to some of the differences seen between sites.

Interestingly, highest morphospecies and phylum richness and coverage characterized Site 4, which also had the most unique morphospecies, and the greatest number of radiolarians and molluscs. Site 4 was closest to Sites 1 and 2 geographically, approximately 170 km off the northernmost coast of Labrador, but water depth (505 m) and frame altitude (16 m) were comparable to Site 1. This site was also one of two with an unobstructed settlement frame placement, which could have increased accessibility to pelagic larvae. The fact that the frame at Site 3 ranked second in richness and first in diversity (see below), highlights that flow (through a frame) may enhance opportunistic encounters. Previous studies have established the importance of food availability, local larval supply, and fine-scale hydrodynamics in larval recruitment and success (*Wahl, 2009*). Potentially, all these variables contributed to richness and abundance of recruits at Site 4.

Site 3 displayed the highest Shannon diversity and highest number of annelid, arthropod, and foraminifer morphospecies. This finding aligns with *Meyer-Kaiser et al. (2019)*, who compared settlement plates deployed at different altitudes (0.25, 0.60, and 0.90 m above bottom) and inferred that annelids drove abundance on plates just above the sea floor. Moreover, Foraminifera msp. 4 and Gammaridea msp. 1 were unique to Site 3, which stood out as the shallowest site (409 m) with an unobstructed frame closest to the bottom (1 m). Nearness to the bottom likely played a critical role, given that Shannon diversity indicated a high number of morphospecies in low abundance. Foraminifers have species-specific preferences or limitations to where they can settle. *Cibicidoides wuellersdorfi* (possibly Foraminifera msp. 1 in the present study) prefers elevation above the bottom (*Lutze & Thiel, 1989*; *Sen Gupta, 2007*) whereas limited dispersal distance characterizes other species (*Sen Gupta, 2007*). Thus, bottom proximity might explain why Foraminifera msp. 4 only occurred at Site 3; conversely, the higher altitude for the frame at Site 4 (11 m) could explain why Foraminifera msp. 2 occurred only there.

Site 3 had the lowest surface coverage, abundance, and density of recruits, supporting that its low altitude potentially subjected it to more frequent visits by benthic predators. Free-living polychaetes at Site 3 (also at Site 1), along with Gammaridea msp. 1, may have been feeding on recruits or dislodging them. Multiple studies document the role of predation in limiting larval recruitment in temperate and tropical regions (*Osman, Whitlatch & Malatesta, 1992*; reviewed in *Jenkins, Marshall & Fraschetti, 2009*). In shallow-water studies in the west Antarctic Peninsula, *Bowden et al. (2006)* reported higher colonization rates in sheltered settlement frames, attributing differences to fewer predators such as errant polychaetes. The interplay of an unobstructed frame (*i.e.*, high encounter rates, fewer sheltered surfaces) and proximity of benthic predators could have contributed to the high Shannon diversity but low-density community of recruits at Site 3.

The highest morphospecies abundance and density occurred at Site 1, which was comparable in frame altitude (11 m) and water depth (499 m) to Site 4 but was mounted flat against the mooring structure (*i.e.*, obstructed). This site was also where the highest abundance of any one morphospecies was found (the stolonate colonial hydrozoan

Campanulariidae msp. 3), suggesting that the mounting method alone was not a major driver of recruitment patterns in all metrics, and that it likely acts in combination with other factors such as proximity to the sea floor. Previous studies documented higher recruitment rates near more sheltered components or sides of settlement plates as a result of fine-scale hydrodynamic fluctuations and protection against predators (*Bowden et al., 2006*; *Lacharité & Metaxas, 2013*); and that some species preferentially position themselves relative to water flow (*Mullineaux & Butman, 1990*; *Meyer-Kaiser et al., 2019*). Although directional positioning relative to water flow over and through the differently mounted frames was not measured in the present study, such preferences may have influenced recruitment to these settlement frames.

Site 2, with the highest frame altitude (60 m) and greatest water depth (960 m), showed the lowest morphospecies and phylum richness, number of unique morphospecies, and surface coverage overall. Annelid and mollusc recruits were absent, and just a single morphospecies of arthropod (Copepoda msp. 1) was documented. The frame altitude and site depth may have limited the number of recruits that could access the substrata, noting that the site also lacked otherwise ubiquitous morphospecies (*e.g.* Campanulariidae msp. 2, *Eudendrium* msp. 1).

Larval dispersal can strongly affect recruitment, often as a function of planktonic larval duration. The gregariousness of eudendriids mentioned by *Wasserthal & Wasserthal (1973)* likely limits their ability to colonize frames well above the bottom and deployed for a relatively short time period. Interestingly, annelids and amphipods occurred at all three shallower sites but not at the deepest (Site 2). This pattern aligns with a study in the Siberian Arctic deep sea, where *Vedenin et al. (2021)* reported annelids and amphipods crowding (*e.g.*, overlapping of upper and lower species' limits) at 400–800 m (~Sites 1, 3, and 4 here), but no crowding from 800 to 1,000 m (~Site 2). Similarly, a study of amphipod diversity as function of depth around Iceland reported species richness peaking at ~500 m before declining (*Lörz et al., 2021*). While it is impossible to tease out these factors and others (*e.g.*, intra-annual variation, small-scale environmental conditions), overall site- and substratum-specific factors clearly affect recruitment in complex, interconnected ways.

## CONCLUSIONS

As the polar seas become increasingly impacted by human infrastructure and pollution, it becomes urgent to understand how deep-sea species in these regions recruit to hard-bottom substrata, particularly across both natural and artificial substratum types. This study showed that early recruitment patterns (<1 y) on hard surfaces in the Labrador Sea result from complex interplays. Substratum characteristics such as material, surficial complexity, and microhabitats affected morphospecies richness, abundance, diversity, and coverage, while factors such as frame altitude, depth, and water flow present at different geographic sites also played a role. High abundance and richness of recruits on plastic and mesh (*i.e.*, a more complex form of plastic) suggests that these anthropogenic materials may provide a highly suitable or even preferred substratum over stone and wood for some recruits, which could have impacts on species distributions as plastics increase in the deep sea. Additionally, colonial hydrozoans, particularly a campanulariid hydrozoan,

dominated across all substratum types and geographic sites, contributing the highest abundance and surface coverage, as well as potentially playing a crucial role in early succession as hosts for epibionts such as foraminifers and caprellid amphipods. Overall, this study provides foundation knowledge of recruitment dynamics at bathyal depths across four differing substratum types in a high-latitude region, while also providing a basis for future work to expand on larval preferences across artificial and natural hard-bottom substrata.

## ACKNOWLEDGEMENTS

The authors extend special thanks to Maria Baker (INDEEP) for supplying the initial settlement frame and plans; to the Ocean Sciences Field Services for construction of subsequent frames; to Emaline Montgomery, Kaitlin Casey, and other members of the Mercier Lab for frame deployment and retrieval; to Shawn Meredyk, Camilla Parzanini, the CCGS *Amundsen* crew, and the ArcticNet and Amundsen Science teams (in 2017, 2018, 2019, 2020) for their help during frame deployments and retrievals on scientific moorings and landers; to David Côté and the Integrated Studies and Ecosystem Characterization of the Labrador Sea Deep Ocean (ISECOLD) project at the Department of Fisheries and Oceans (DFO) Canada; and to Bárbara de Moura Neves and Paul Snelgrove for feedback during project and manuscript development.

### Funding

Funding was provided by the Natural Sciences and Engineering Research Council of Canada (NSERC) through Discovery and Ship-time grants to Annie Mercier. There was no additional external funding received for this study. The funders had no role in study design, data collection and analysis, decision to publish, or preparation of the manuscript.

### Grant Disclosures

The following grant information was disclosed by the authors:
Natural Sciences and Engineering Research Council of Canada (NSERC).

### Competing Interests

The authors declare that they have no competing interests.

### Author Contributions

- Sophie Wolvin conceived and designed the experiments, performed the experiments, analyzed the data, prepared figures and/or tables, authored or reviewed drafts of the article, and approved the final draft.
- Jean-François Hamel conceived and designed the experiments, authored or reviewed drafts of the article, project administration, sample collection, and approved the final draft.
- Annie Mercier conceived and designed the experiments, authored or reviewed drafts of the article, project administration, resources, supervision, and approved the final draft.

## Data Availability

Raw data is available in the Supplemental Files.

## Supplemental Information

Supplemental information for this article can be found online at http://dx.doi.org/10.7717/peerj.19850#supplemental-information.

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
