# Peer review of "Between a rock and a hard place: experimental assessment of recruitment patterns in a bathyal environment of the Low Arctic"

_PeerJ, doi:10.7717/peerj.19850_

## Round 0.1 · original submission · Major Revisions

· Academic Editor

Major Revisions

Both reviewers appreciated the research in this paper and have recommended some suggestions that I think will improve the manuscript. Both indicate areas around clarity and ties to the literature in particular, as well as some introductory material around the importance of this system that would help.

Reviewer 1 ·

Basic reporting

This manuscript reports on a recruitment study in the bathyal Labrador Sea. The experimental design is clear, and the conclusions are supported by the data. I have only minor suggestions for improvement of the manuscript.
One question is why the morphospecies were not identified using molecular techniques (barcoding). All the samples were preserved in ethanol, so it should have been possible. It would be interesting, particularly for some of the more common taxa, to have species-level identifications. Did the authors deem molecular identification impractical, given the dearth of reference sequences for deep-sea or Arctic fauna?
The introduction has a heavy emphasis on methodological or procedural choices in recruitment studies. I suggest revising this section to focus more on why recruitment varies between substrates. This would make the introduction section more conceptual and less procedural. It’s not just that different species prefer different microhabitats – it’s that different microhabitats have different small-scale hydrodynamics, predation risk, and competitive advantages. Some key papers to read and potentially cite are:

Mullineaux LS, Garland ED (1993) Larval recruitment in response to manipulated field flows. Mar Biol 116:667–683.
Walters LJ, Wethey DS (1996) Settlement and early post-settlement survival of sessile marine invertebrates on topographically complex surfaces: The importance of refuge dimensions and adult morphology. Mar Ecol Prog Ser 137:161–171.
Walters LJ, Wethey DS (1986) Surface topography influences competitive hierarchies on marine hard substrata: a field experiment. Biol Bull 170:441–449.

The paragraphs beginning at line 80 and line 101, respectively, relay in fine detail the experimental designs and findings of other researchers. I suggest shortening these paragraphs and better integrating the concepts into the broader discussion of recruitment, rather than focusing on the procedural components of previous experiments.
The introduction should finish by defining some key questions or hypotheses that the study aims to answer. A general direction is defined, but this should be made more specific. Given the structure of the discussion, I suggest that the authors could list two questions: “How does recruitment differ between substrata of different complexities?” and “How does recruitment vary between study sites?” These questions are answered very well in the results and discussion; they should just be named in the introduction to focus the study.

Experimental design

The methods are straightforward and described sufficiently.
Line 284: Is Foraminifera a phylum?
The results are very clearly written and easy to follow. Nicely done.

Validity of the findings

Line 418: Check the accuracy of that citation.
Line 428: Interesting hypothesis, that gastropods lay eggs on hydroids so their juveniles have a food source right there! Has this ever been demonstrated at shallower depths?
Line 433: Excellent summary of this discussion paragraph.
Line 463: Yes, excellent point, but there is also a great paper you could look at:
Pinochet J, Urbina MA, Lagos ME (2020) Marine invertebrate larvae love plastics: Habitat selection and settlement on artificial substrates. Environ Pollut 257:113571.

The discussion does a really nice job of comparing the results of the present study to previous work.
Lines 533-537: This is a run-on sentence. Please split it into two sentences.
It seems from the discussion that you are concluding the differences in recruitment between sites were mostly driven by differences in altitude and orientation of the panels. Are there any environmental factors that varied strongly between sites that could explain these differences? If the environmental factors are largely similar, then this is a good conclusion!
The conclusion largely summarizes the results and recapitulates the abstract. I suggest reshaping this section to highlight the broader implications of your study. What is the one take-home message?
Overall, this is a very clear and well-presented manuscript.

Additional comments

This is a good manuscript that only needs minor revision.

Reviewer 2 ·

Basic reporting

This is a well written paper with ample background context (as much as there can be for polar deep sea recruitment!). The author’s figures are well made and for the most part clear (see notes on a few). The structure is professional but could use a bit of clarity, especially in the results and discussion. There is actually a bit too much data reported without the accompanying synthesis. Authors should substantially cut back the amount of information reported (especially in the results) and focus in on their major findings and synthesize what is most important. As of now, it reads much more like an initial lab report – sharing every possible combination of data but lacking big picture assessments.

Experimental design

This research is defined and relevant, filling a knowledge gap about recruitment and community assemblage across substrate types at high latitude and depth. They described their methods reasonably and clearly, although with the opportunistic design of the study, could be difficult to replicate. Further, while I appreciate the limitations of research of this nature, I feel some of the assessments are overstated. With only a single mosaic deployed at each site, each at different depths (and different altitudes above the seafloor), I am not sure any statistical assessment between sites is warranted (or assessment of differences between years/depths).

Validity of the findings

The author’s findings are quite clearly assessed to the fullest. They have provided all relevant background data. I question the robustness and statistical soundness of comparing samples from disparate sampling locations (as noted above), but understand the limitation in such a region. The authors should be sure to acknowledge this caveat more clearly. The results, while valid, are too long and read like a list which is difficult to follow. The authors did a good job relating their findings to other work throughout the discussion, especially as they related to specific taxa. The conclusions summarize the work well, but the authors should consider adding a few sentences more about why we should care about these differences, especially in high altitude places (e.g. increased shipping – increased infrastructure that will be made from some of these materials, changing population connectivity?)

Additional comments

Minor comments:

Abstract
17 – “better understand”
29-32 – This sentence feels overstated for your actual results about these patterns. I would leave it out of the abstract and just note that other factors such as location, depth, and altitude above sea floor may have also played a role. But you should be focusing in on the main point that you were testing different substrates.
53- remove “more or less extended” and cite something here

Introduction
105- write out years
115-125 – add a “why” rather than just describing the novelty of the method.
120 – four sites?
124 – I didn’t notice an assessment of depth or between year recruitment patterns in your results, and the years in particular are only reported in supplementary. I wouldn’t include either here.

Methods
Either here or in the introduction, add a paragraph about your study system (what do we know about deep sea benthic taxa in that region?)
216 – is there something you can cite to describe why you did this?
220 – for taxa overlapping two locations/habitats, what did you do? And how did you determine initial recruitment location if this was the case?
226 – do I read this right that the % occurrence is actually presence/absence frequency in a certain location (on the block) – so if sp 1. Is found on four faces and on three of those faces they were found in a certain location so sp. 1percent occurrence for that location would be 75%? If so, I think you can make this a little bit clearer – and may be what is leading to my confusion in some of your tables/figures. Also, curious how/why this is the relevant metric (can you cite this from somewhere else maybe?)

Results
256 – comma in 127 724 “127,724” – and everywhere else in results you have these large numbers (e.g. lines 260, 261, 276 , 281, 291 etc.).
259 – remove “was” or rephrase
260 – I don’t believe you mean in addition to (as well as) – instead “17 of which were sessile (127,451 ind) and 11 of which were motile (165 ind).”
262- be consistent – either mspp or morphospecies throughout
289 – Start this section with a variation of the sentence currently on 294. Saying They differed substantially and then give the details. I also think much of this section could just be in tables. Most of this section could also be incorporated into tables/figures, only reporting the overall findings in text (total abund. Different amongst substrata, plastic the most, wood the least) and the rest in a table or values reported on the associated figures
297 – mesh vs stone?
309 – Are you saying that the exact same set of taxa colonized these two? If not, I would change the structure of this sentence.
320 – Fig. 4C.
321 – 330 - I was working backwards a bit, so refer to my next comment – I think there is maybe a more intuitive way to report this – especially because the %s are really hard to follow and as is it doesn’t give intuitive information about how many taxa chose multiple substrates/microhabitats etc.

343-347 – Based on the numbers it appears there is overlap in who settled where, so these percentages don’t feel useful (since they add up to >100%). I’m honestly not sure what this assessment is telling me. There were 10 motile morphospecies – nine of them used sheltered microhabitats (90%) and five of them (four of the nine and the tenth?) also used unsheltered microhabitats (why is this 56% and not 50%?). Would it be clearer to report in the overlaps? “Of the 10 motile species five only occurred in sheltered habitats, four occurred in both sheltered and unsheltered, and one occurred only in unsheltered.” This feels much clearer.

378-387 – I am not sure what the scientific reasoning behind assessing colonial vs.unitary and sessile vs motile morphospecies at both the site and block levels. I would just report this for one or the other, especially since going down to the substratum level didn’t change your findings.

While I appreciate how thorough you’ve been in reporting so many values, the results section is hard to get through – chunky and lacking flow. There are far too many parentheticals and directly reported values. I have suggested a few possible places to slim it down. Generally, I suggest putting more summary information into tables and/or figures and reporting less numerical information in text. Further, focus on relevant findings for in-text with reference to the figures/tables for further information.

Figure 7 – I don’t know that I understand the “percentage of the total number of occurrences.” I like the overall setup of the heat map – that part is intuitive, but I can’t tell what ¬the numbers actually mean (even now after re-reading the methods). I am not seeing anything sum to 100. Also why are there sometimes zeros but not always? Again, the figure looks good but it needs a bit more explanation.¬

Table 3 – Same problem as Figure 7. I don’t understand what you mean by percentage of occurrences. Wouldn’t that mean they would need to sum to 100%? I am unsure how a single taxa can have 50% in unsheltered and 100% unsheltered at a given site/substratum combo. Still – even after re-reading the methods, I don’t think this is intuitive –at minimum, reexplain/state the metric here.

Discussion/Conclusion

418 – Kristen-Meyer should be Meyer-Kaiser et al. (2022)
469-474 – a bit awkward wording here, also why is the acrylic similar to wood?
509 – Did Meyer-Kaiser et al. 2019 have sites in eastern Greenland and western Svalbard portions of the Fram? I thought it was just one frame. Maybe just refer to the Fram without the qualifiers – it sounds like you’re referencing a different region than before

Overall in pretty good shape, but I’d encourage you to relate it more to the actual study ecosystem rather than focus so heavily on methods/recruitment comparisons in other regions. How does this knowledge mesh with (or not) our current understanding of the Labrador sea region? (More info like you reported in line 425!)

---

## Round 0.2 · accepted · Accept

· Academic Editor

Accept

Thanks for thoroughly addressing all of the comments from the two reviewers. The new manuscript is improved and easier to read. Thanks and congrats!

Reviewer 2 ·

Basic reporting

I commend the authors on their thorough consideration of the comments of both reviewers. I especially appreciate their clarifying text about the method of calculating percentages and their further streamlining of the results section – it reads much smoother now. The manuscript has been greatly improved and I recommend it for publication with only a handful of copy-editing level edits. The authors’ reporting is clear and unambiguous with thorough review of literature and appropriate analyses. Their findings are novel and overall an exciting addition to our understanding of recruitment dynamics in both high latitude and deep sea environments.
(very) minor edits:
315-316, 353: a few extra spaces before commas/periods
629: spell out year (?)
Center spelled two different ways across figs/tables

Experimental design

no comment

Validity of the findings

no comment